# INR-V: A Continuous Representation Space for Video-based Generative Tasks

**Bipasha Sen**[*]                                                         *bipasha.sen@research.iiit.ac.in*
*IIIT Hyderabad*

**Aditya Agarwal**[*]                                                      *aditya.ag@research.iiit.ac.in*
*IIIT Hyderabad*

**Vinay P Namboodiri**                                                    *vpn22@bath.ac.uk*
*University of Bath*

**C. V. Jawahar**                                                         *jawahar@iiit.ac.in*
*IIIT Hyderabad*

**Reviewed on OpenReview:** *https://openreview.net/forum?id=aIoEkwc2oB*

## Abstract

Generating videos is a complex task that is accomplished by generating a set of temporally coherent images frame-by-frame. This limits the expressivity of videos to only image-based operations on the individual video frames needing network designs to obtain temporally coherent trajectories in the underlying image space. We propose INR-V, a video representation network that learns a continuous space for video-based generative tasks. INR-V parameterizes videos using implicit neural representations (INRs), a multi-layered perceptron that predicts an RGB value for each input pixel location of the video. The INR is predicted using a meta-network which is a hypernetwork trained on neural representations of multiple video instances. Later, the meta-network can be sampled to generate diverse novel videos enabling many downstream video-based generative tasks. Interestingly, we find that conditional regularization and progressive weight initialization play a crucial role in obtaining INR-V. The representation space learned by INR-V is more expressive than an image space showcasing many interesting properties not possible with the existing works. For instance, INR-V can smoothly interpolate intermediate videos between known video instances (such as intermediate identities, expressions, and poses in face videos). It can also in-paint missing portions in videos to recover temporally coherent full videos. In this work, we evaluate the space learned by INR-V on diverse generative tasks such as video interpolation, novel video generation, video inversion, and video inpainting against the existing baselines. INR-V significantly outperforms the baselines on several of these demonstrated tasks, clearly showcasing the potential of the proposed representation space.

## 1 Introduction

Learning to generate complex spatio-temporal videos from simple distributions is a challenging problem in computer vision that has been recently addressed in various ways Tian et al. (2021); Tulyakov et al. (2017); Clark et al. (2019); Skorokhodov et al. (2021); Ding et al. (2019); Yu et al. (2022); Yan et al. (2021). State-of-the-art (SOTA) works Skorokhodov et al. (2021); Tian et al. (2021); Yu et al. (2022) treat video generation as a task of generating a sequence of temporally coherent frames. Although such networks have advanced the SOTA to generate high-quality frames (such as carefully crafted eyes, nose, and mouth for talking-head

---

[*]Equal contribution.

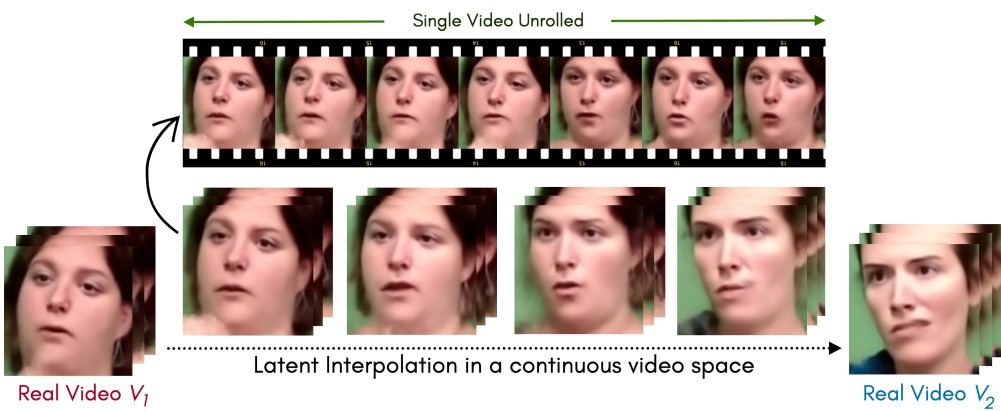

Figure 1: Demonstrating the continuity of the video space learned by INR-V by interpolating novel videos between two real videos $V_1$ and $V_2$. Note that content (identity, hair) and motion (pose, expressions) gradually transition as we traverse the trajectory in the latent space between $V_1$ and $V_2$'s latents.

videos), they come with a major limitation: They rely on an image space. This limits the application of the learned space to image-based operations such as animating images and editing on frames. Direct operations on videos, such as interpolating intermediate videos between two videos and generating future segment of a video, become difficult. This is because such operations require learning the set of frame and motion constraints and ensuring that they are coherently learned.

We propose that videos can be represented as a single unit instead of being broken into a sequence of images. One can learn a latent space where each latent point represents a complete video. However, with existing video generator architectures, such representations are difficult. Firstly, such a video generator would be made of several 3D convolution operations. As the dimension and length of the video increase, such an architecture would become drastically computationally expensive (a GPU with limited memory can only fit a video of limited dimension). Secondly, videos are high-dimensional signals spanning both spatial and temporal directions. Representing such a highly expressive signal by a single latent point would require complicated generator architectures and a very high-dimensional latent space. Instead, videos can be parameterized as a function of space and time using implicit neural representations (INRs). Any point in a video $V_{hwt}$ can be represented by a function $f_\theta(h, w, t) \rightarrow RGB_{hwt}$ where $t$ denotes the $t^{th}$ frame in the video and $h$, $w$ denote the spatial location in the frame and $RGB$ denotes the color at the pixel position $\{h, w, t\}$. Here, the dynamic dimension of videos (a few million pixels) is reduced to a constant number of weights $\theta$ (a few thousand) required for the parameterization. A network can then be used to learn a prior over videos in this parameterized space. This

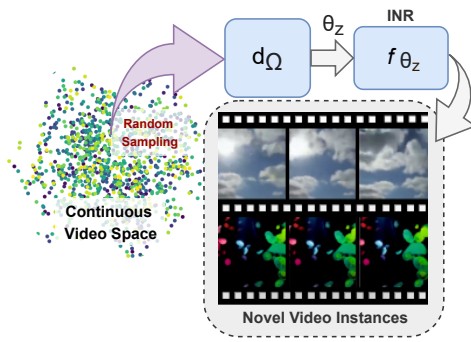

Figure 2: **Overview of INR-V:** INR-V learns a continuous video space by first parameterizing videos as implicit neural representations denoted by $f_{\theta_z}$, where $z$ denotes a unique video instance $V_z$. Next, a meta-network based on hypernetworks denoted by $d_\Omega$ is used to learn a continuous representation over the neural representations. $d_\Omega$ is conditioned by an underlying continuous video space where each point denotes the condition for a complete video.

can be obtained through a meta-network that learns a function to map from a latent space to a reduced parameter space that maps to a video. A complete video is thus represented as a single latent point.

We propose INR-V, a video generator network with a continuous video representation space based on learning an implicit neural representation for videos. It is illustrated in Fig. 2. INR-V is made of key elements that, when combined, makes it ideal for video representation: (1) Its INR is free of expensive convolutional layers (millions of parameters) such as in the existing architectures Tian et al. (2021); Skorokhodov et al. (2021)

and relies on a few layers of traditional multi-layered perceptrons (MLPs), leading to a very few parameters (a few thousand). (2) Having very few parameters, INR's weights can be populated using a secondary meta-network called hypernetwork Ha et al. (2016) that learns a continuous function over the INRs by getting trained on multiple video instances. (3) It is trained on a deterministic distance loss, such as Euclidean or Manhattan distance. This allows INR-V to learn the exact requirements of a coherent video directly from the ground truth video instances.

Hypernetworks have seen wide applications in graphics Sitzmann et al. (2020b; 2021); Chiang et al. (2021); Sitzmann et al. (2019); however, they have been seldom used for videos. Hypernetworks are notoriously unstable to train, especially on the parameterizations of highly expressive signals like videos. Thus, we propose a key prior regularization and a progressive weight initialization scheme to stabilize the hypernetwork training allowing it to scale quickly to more than 30,000 videos. As we show in the experimental section, INR-V demonstrates an expressive and continuous video space by getting trained on these datapoints. The learned prior enables several downstream tasks such as novel video generation, video inversion, future segment prediction, and video inpainting directly at the video level. As shown in Fig. 1, INR-V also showcases smooth interpolation of novel videos between two videos by traversing the path between their latent points. Interpolation morphs different identities and motions and generates coherent videos. Interestingly, the properties demonstrated in this work are not enforced at training but are natural outcomes of the continuous video space. To summarize, our contributions in this work are as follows:

1. We propose considering videos as a single unit and learning a continuous latent space for videos where each latent point represents a complete video.

2. We propose INR-V, a video representation technique that parameterizes videos using INRs, bringing down the dimension of a video from a dynamic few million to a constant few thousand. INR-V uses a hypernetwork as a meta-network to learn a continuous space over these parameterizations.

3. We demonstrate the benefit of a key regularization and progressive weight initialization scheme to stabilize the hypernetwork training. We scale the hypernetworks to more than 30,000 video points enabling it to learn a continuous meaningful latent space over the INRs.

4. Lastly, we demonstrate key properties of the learned video space, such as video interpolation, video inversion, and so on, by conducting several experiments and evaluations.

## 2 Related Work

**Video Generation.** Video generation aims to produce novel videos from scratch. It falls under the paradigm of 'video synthesis' that encompasses several categories, including (1) Video prediction Luc et al. (2020); Moing et al. (2021); Walker et al. (2021): that predicts the next set of frames given the current frames, (2) Frame interpolation Park et al. (2021); Niklaus & Liu (2020); Niklaus et al. (2017); Zhang et al. (2021): that interpolates frames between given frames of a video. These tasks generate the unseen portion of the video based on the context of the seen portion. On the other hand, video generation produces videos without any expressive prior context, making the task more challenging. The complexity of the problem has led to a plethora of works in this area Tian et al. (2021); Tulyakov et al. (2017); Skorokhodov et al. (2021); Clark et al. (2019); Ding et al. (2019); Yu et al. (2022). VideoGPT Yan et al. (2021) tackled this challenge by first reducing the raw videos of up to $128 \times 128$ dimension to a quantized space. It then trained a transformer architecture to model a prior over the quantized space. Our architecture is conceptually similar to VideoGPT, which used a likelihood-based generative model to learn a video prior. However, VideoGPT operates on a quantized space that is discontinuous, making the prior less expressive. INR-V, on the other hand, models a continuous video space. VideoGPT also consists of 3D convolution layers making the model computationally expensive for larger videos. INR-V is a simple MLP, based on a continuous parameterization scheme of INRs, making it agnostic to the video dimension. This allows scaling to multiple resolutions ($64 \times 64$ or $256 \times 256$) at inference without any architectural changes or finetuning. More recent works StyleGAN-V Skorokhodov et al. (2021), DIGAN Yu et al. (2022), and MoCoGAN-HD Tian et al. (2021) are a GAN-based setup that model videos as a temporally coherent trajectory over an image space. Use of a continuous representation

space for videos has been considered before in Fernando et al. (2015); Bilen et al. (2016) for the task of action classification. However, in this work, we focus on learning a representation space for generative tasks.

**Hypernetworks.** Hypernetworks Ha et al. (2016) were introduced as a metafunction that initializes the weights for a different network called the primary network. Hypernetworks have been widely used for several purposes, starting from representation learning for continuous signals Park et al. (2019); Sitzmann et al. (2021; 2020a;b); Mescheder et al. (2018); Sitzmann et al. (2019), compression Nguyen et al. (2022); Gao et al. (2021), few-shot learning Sendera et al. (2022); Lamb et al. (2021), continual learning von Oswald et al. (2019), language modeling Suarez (2017). We use hypernetworks to populate our primary video generation network, an MLP parameterizing different video instances.

**Implicit Neural Representations.** In this paradigm, a continuous signal is represented by a neural network. INRs have had wide adaptations in 3D Computer Vision Park et al. (2019); Genova et al. (2019); Sitzmann et al. (2018); Mescheder et al. (2018); Sitzmann et al. (2021); Mildenhall et al. (2020) and Computer Graphics Guo et al. (2021); Yao et al. (2022). Recently, INR was adopted for images Skorokhodov et al. (2020) and videos Chen et al. (2021); Sitzmann et al. (2020b); Yu et al. (2022); Chen et al. (2022). INR-GAN Skorokhodov et al. (2020) first showed the application of INRs in generating high-quality images by replacing the generator component of StyleGAN2 Karras et al. (2019) with an MLP-based INR. It then used a hypernetwork to populate the INR. Unlike INR-GAN, which is trained using a stochastic discriminator, INR-V relies on a deterministic distance-based loss to train the hypernetwork. SIREN Sitzmann et al. (2020b) proposed periodic activation functions for INRs as a replacement for ReLU activation to parameterize many different data types like images, videos, sounds, and 3D shapes, with fine details. NeRV Chen et al. (2021) designed an implicit function as a continuous function of time and used convolution blocks at each time step to parameterize discrete frames showcasing an improved frame quality over SIREN. Recently, VideoINR Chen et al. (2022) was proposed that used INRs for video superresolution. DIGAN Yu et al. (2022) incorporated INRs made of MLP layers for video generation. It consisted of two separate networks that generated spatial and temporal codes for generating videos in a frame-wise fashion. StyleGAN-V Skorokhodov et al. (2021) also incorporated INRs and relied on continuous non-periodic positional encodings for each timestep of a video. Like NeRV, StyleGAN-V used traditional convolution operations for frame-by-frame video generation. Both DIGAN and StyleGAN-V used a GAN setup to train the video generators. INR-V is based on MLPs with ReLU activation trained in a fashion similar to Light Field Networks (LFNs) Sitzmann et al. (2021). LFNs proposed a novel neural scene representation for novel view synthesis and trained a hypernetwork over multiple object instances using distance-based losses like Euclidean or Manhattan distance. Like LFNs and INR-GAN, INR-V parameterizes the entire signal (a video) using INRs and relies on a single hypernetwork to generate the INRs. However, unlike LFNs and INR-GAN, INR-V encodes a denser representation of a volumetric 3D signal $\in \mathbb{R}^3$ data making hypernetwork training more challenging.

## 3   INR-V: Implicit Neural Representation for Video Synthesis

Each video instance $V_n$ consists of pixels at locations $(h, w)$ at $t^{th}$ frame. We have a particular parameter vector $\theta_n$ that is used by a network $f$ to generate the value of the color $RGB_{hwt}$ for that pixel location $(h, w, t)$. We need to learn a network $d$ with parameters $\Omega$ that predicts the parameters $\theta_n$ for a particular video $V_n$. Here, $d$ is a hyper-network. The overall approach to train the network is illustrated in Fig. 3.

### 3.1   Hypernetwork for Modeling Multiple Video Instances

As $f_\theta$ implicitly stores a single video signal, any new video would need its own implicit function. Let $f_{\theta_n}$ denote the implicit function for a given video $\{V_n\}_{n=1}^N$ where $N$ is the total number of available videos in the training dataset $\mathcal{D}$. Each of these implicit functions, $f_{\theta_n}$ can be modeled using a neural network trained on each pixel value of the video $V_n$. Thus, implicit functions minimize the following objective:

$$L(\theta_n) = \frac{1}{T} \frac{1}{W} \frac{1}{H} \sum_{t=1}^{T} \sum_{w=1}^{W} \sum_{h=1}^{H} (f_{\theta_n}(h, w, t) - RGB_{hwt})^2 \tag{1}$$

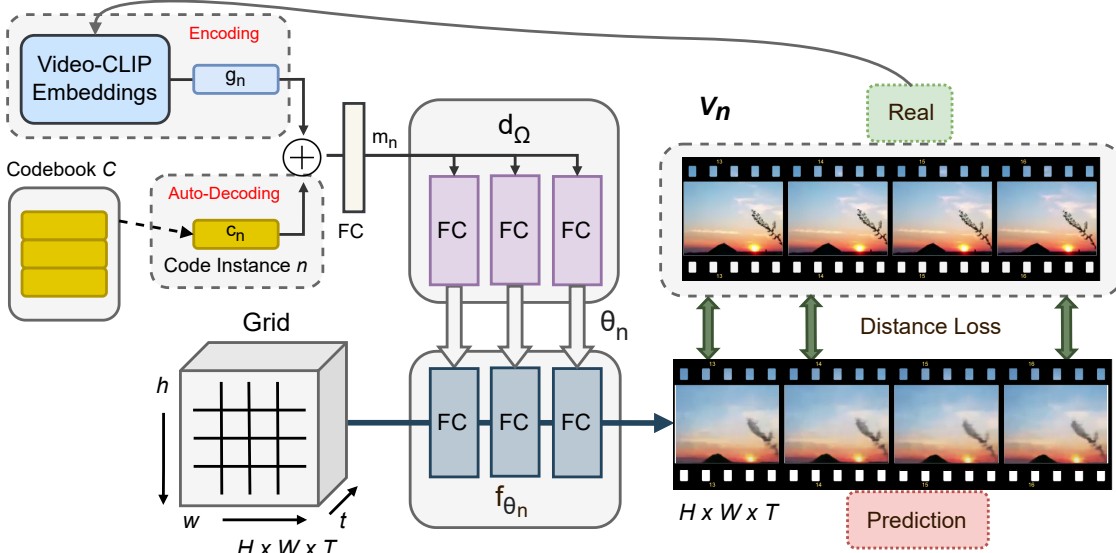

Figure 3: **Architecture of INR-V:** Any video instance $V_n$ is represented by its corresponding implicit neural representation, an MLP, $f_{\theta_n}$. $f_{\theta_n}$ takes a grid as input denoting the pixel positions of the video encoded using periodic positional encodings. It then generates the pixel values for all the positions. $f_{\theta_n}$ is initialized by a meta-network called hypernetworks denoted as $d_\Omega$ composed of a set of MLPs. $d_\Omega$ is conditioned by an instance code $m_n$ unique to every video instance $V_n$. $m_n$ is trained by combining (1) auto-decoding framework to regress to a code $c_n$ and (2) encoding-framework to regularize the space using CLIP embedding that generates $V_n$'s semantic code $g_n$. At the time of inference, $m_n$ is randomly sampled from an underlying learned distribution $\tau$.

Generating a novel video $V_z$ translates to generating a novel implicit function $f_{\theta_z}$ that represents the video meaningfully. Let us consider $f_{\theta_z}$, an unseen sample from an underlying distribution $\Phi$. Each point in the distribution $\Phi$ denotes an implicit function of a meaningful video. To randomly sample $f_{\theta_z}$, we make use of a meta-network to learn the distribution $\Phi$.

We use a hypernetwork $d_\Omega$ as a meta-network to parameterize $f_\theta$, such that $d_\Omega(m_n) = \theta_n$ for video instance $V_n$. Here $m_n$ is a a $d$-dimensional point in the latent space, say $\tau$, and serves as an instance code for $V_n$. Given enough number of samples $N$, $d_\Omega$ learns to map the latent codes sampled from $\tau$ to their corresponding parameterized space $\Phi$, as shown in Fig. 3. The parameters $\theta_n$ are then used to initialize $f$ to generate $V_n$.

Let us consider $\tau$ as a meta-distribution such that $\{m\}_\mathcal{D} \in \tau$. At the time of inference, $m_z$ can be sampled from $\tau$. As $d_\Omega$ has learned a valid representation over $\Phi$, $m_z$ enables $d_\Omega$ to generate a meaningful implicit function $f_{\theta_z} \in \Phi$. Sampling from $\tau$ can be made straight forward by making sure $\tau$ is regularized during training. At the time of training, $\Omega$ and $\{m_n\}_{n=1}^N$ are optimized together. $\theta$ is a non-learnable parameter and $f$ is initialized as the output of $d_\Omega$. The following objective is optimized:

$$L(\Omega, m) = \frac{1}{N}\frac{1}{T}\frac{1}{W}\frac{1}{H} \sum_{n=1}^{N} \left( \sum_{t=1}^{T} \sum_{w=1}^{W} \sum_{h=1}^{H} (f_{\theta_n}(h, w, t) - RGB_{ijk})^2 \right) \quad \text{and} \quad \theta_n = h_\Omega(m_n) \qquad (2)$$

### 3.2 Regularizing $\tau$ for Hypernetwork Conditioning

To generate a novel video, a random latent $m_z$ is sampled from the latent space $\tau$. $d_\Omega$ is then conditioned on $m_z$ generating an implicit function $f_{\theta_z} \in \Phi$. In a standard hypernetwork training Sitzmann et al. (2019; 2021); Park et al. (2019); Sitzmann et al. (2020b), $m_n$ is optimized in an auto-decoding framework as given in Eqn. 2. However, given the complexity of the signal $V$ (a 3D volumetric representation) that $d_\Omega$ has to model, $\{m\}_\mathcal{D}$ can collapse to a single point if $\tau$ is not regularized at the time of training, bringing the expressiveness

of $d_\Omega$ down to a single implicit function. We regularize $\tau$ by leveraging pretrained CLIP Radford et al. (2021) designed for generating semantically meaningful embeddings for images. We design Video-CLIP that encodes an entire video $V_n$ to a vector $g_n$. As shown in Fig. 4, Video-CLIP first generates the image-level CLIP embeddings. These embeddings are then passed through a bi-directional GRU. The mean of the hidden state outputs of the final layer produces $g_n$. As shown in Fig. 3, the regularized instance code $m_n$ is now:

$$m_n = \phi(c_n, g_n) \tag{3}$$

where $c_n$ is the instance code of $V_n$ optimized in an auto-decoding fashion at the time of training, and $\phi$ is a neural network. The pretrained CLIP embeddings are kept frozen during training and the learnable parameters are the instance codes $c_n$ that are regularized by $g_n$. CLIP regularization encourages the latent codes to be spaced sufficiently apart by leveraging predefined semantic encoding. This helps avoid mode collapse during the initial stages of training. Please find the ablation on CLIP regularization in Appendix A.1.1.

### 3.3 Progressive Training

A video is a dense 3D volume mandating its neural representation to model every single point in the volume. Although implicit representations have a constant number of parameters made of only a few layers of MLPs in our case, learning a meta-function using a hypernetwork over such dense representations is challenging. As a result, if not appropriately initialized, the hypernetworks can easily collapse to a single representation despite CLIP regularization. Moreover, a sub-optimal hypernetwork initialization could result in a significantly longer convergence period. To tackle this challenge, we adopt a progressive initialization scheme. Firstly, the training is divided into multiple stages. Each stage, denoted by $\{l\}_{l=1}^{\mathcal{K}}$ where $\mathcal{K}$ is the total number of stages, is made of a subset of the training dataset $\mathcal{D}$. The number of samples $N_l$ in each stage $l$ is given as:

Figure 4: **Video-CLIP**: Encoding a video $V_n$ to a latent vector $g_n$ by using image-level CLIP encodings.

$$N_l = \begin{cases} N_{l-1} + \epsilon_l & l > 1 \\ \mathcal{C} & l = 1 \end{cases} \tag{4}$$

where $\mathcal{C}$ is a constant and $\epsilon_l$ denotes the number of additional samples for $l^{\text{th}}$ stage. Each step $l$ consists of $\{V_n\}_{n=1}^{N_l}$ datapoints that is computed as:

$$\{V_n\}_{n=1}^{N_l} = \{V_i\}_{i=1}^{N_{l-1}} + \{V_j\}_{j=N_{l-1}}^{N_{l-1}+\epsilon_l} \tag{5}$$

where the order of set $\{V\}$ is maintained across the training stages. At the start of the training, the model is trained on $\mathcal{C} < 10$ examples. This allows the hypernetwork to quickly adapt to the handful of examples and initialize the weights. However, jumping from $\mathcal{C}$ to $\sim 30{,}000$ samples causes the network to collapse again. Thus, we adapt the network progressively to the given examples. Each stage of the progressive training is a full training of the model, with the weights in the current stage initialized with the weights learned from the previous stage. This includes reusing the instance codes $c_n$ learned at a previous stage $l-1$ in the current stage $l$. This step is crucial, as without this, the hypernetwork is pushed to re-learn all the instance codes. The new instance codes added in the current stage are initialized from a Gaussian distribution.

## 4 Experiments

**Experimental Setup:** We perform our experiments on (1) How2Sign-Faces Duarte et al. (2020), (2) SkyTimelapse Xiong et al. (2017), (3) Moving-MNIST Srivastava et al. (2015), and (4) Rainbow-Jelly Skorokhodov et al. (2021). Real video samples of each dataset are visualized in Appendix Fig. 19. How2Sign Duarte et al. (2020) is a full-body sign-language dataset consisting of 11 signers. The signers have elaborate facial expressions, mouth, and head movements. We modify How2Sign to How2Sign-Faces by cropping the face region out of all the videos and randomly sample 10,000 talking head videos, each of

| Dataset | Single-INR | | | INR-V | | | | |
|---|---|---|---|---|---|---|---|---|
| | $\mathcal{E} \downarrow$ | $\text{PSNR}_{50} \uparrow$ | $\text{SSIM}_{50} \uparrow$ | $\mathcal{E}_{50} \downarrow$ | $\text{PSNR}_{50} \uparrow$ | $\text{SSIM}_{50} \uparrow$ | $\text{PSNR}_{\text{FULL}} \uparrow$ | $\text{SSIM}_{\text{FULL}} \uparrow$ |
| How2Sign-Faces | 4.83 | 29.72 | 0.925 | 8.29 | 25.69 | 0.850 | 25.84 | 0.869 |
| SkyTimelapse | 4.69 | 36.19 | 0.943 | 5.87 | 33.69 | 0.931 | 33.94 | 0.924 |
| Moving-MNIST | 3.57 | 37.26 | 0.978 | 6.06 | 29.81 | 0.949 | 29.54 | 0.975 |
| RainbowJelly | 4.17 | 35.93 | 0.918 | 5.02 | 33.34 | 0.937 | 33.57 | 0.938 |

Table 1: Quantitative metrics on reconstruction quality. Comparison set is made of 50 videos per training dataset. INRs trained individually for each video is denoted as Single-INR. INR-V trains a single hypernetwork $d_\Omega$ to populate the INRs of all the videos in the training dataset. $\text{PSNR}_{50}$ and $\text{SSIM}_{50}$ are computed on the comparison set, $\text{PSNR}_{\text{FULL}}$ and $\text{SSIM}_{\text{FULL}}$ are computed on the entire training set. $\mathcal{E}$ is computed on videos with pixel range $[0, 255]$. INR-V performs comparably with Single-INR despite getting trained on huge datasets of more than 30,000 videos.

| Method | How2Sign-Faces | SkyTimelapse | Moving-MNIST | RainbowJelly |
|---|---|---|---|---|
| MoCoGAN-HD | 396.53 | 321.44 | 296.95 | 1856.21 |
| DIGAN | 165.89 | 135.60 | 144.97 | 408.19 |
| StyleGAN-V | 94.64 | **85.05** | 109.85 | 1227.70 |
| INR-V | 161.68 | 153.42 | 103.24 | **260.72** |
| + Denoising | **87.22** | - | **47.28** | - |

Table 2: $\text{FVD}_{16}$ metrics computed on random videos generated by the respective models.

at least 25 frames, of dimension $128 \times 128$. SkyTimelapse Xiong et al. (2017) consists of scenic videos of sky changes. It is made of 1803 videos, each at least 25 frames. The videos are first center-cropped to $360 \times 360$ from an original dimension of $360 \times 620$ and then resized to $128 \times 128$ for training. Moving-MNIST Srivastava et al. (2015) is a video dataset of moving MNIST digits containing a total of 10,000 datapoints. Each video is 20 frames long. RainbowJelly is a single underwater video capturing colorful jellyfishes. The video is first extracted into frames which are then divided into videos of 25 frames each, making a total of 34,526 videos. Similar to SkyTimelapse, the videos are first center cropped to $360 \times 360$ and then resized to $128 \times 128$.

All experiments are performed on 2 NVIDIA-GTX 2080-ti GPUs with 12 GB memory each. All models, except INR-V, are trained at a resolution of $128 \times 128$. To make training computationally efficient, INR-V is trained on a lower resolution of $100 \times 100$ videos. Based on INRs, INR-V can infer directly at multiple resolutions (please refer section 5.2). For evaluations and comparisons, INR-V is inferred at $128 \times 128$ like the other models. The training setup and model architecture are explained in Appendix A.2[1].

## 4.1 Comparing INR-V with Single-INR

INR-V uses hypernetworks to learn a distribution over the INRs of videos. A single hypernetwork $d_\Omega$ can initialize the INRs for multiple videos $\{V_n\}$ based on their respective instance codes $m_n$. Thus, measuring if $d_\Omega$ generates the INR functions $f_\theta$ accurately is crucial. We evaluate this using a set of 50 randomly sampled videos from the training dataset. Each video is first trained to fit a single INR function $f_{\theta_n}$ using Eqn. 1 denoted as Single-INR. Next, the INRs of these 50 videos are populated using a pretrained hypernetwork $d_\Omega$ trained on the entire dataset. We measure the reconstruction quality with PSNR (Peak Signal to Noise Ratio), SSIM (Structural SIMilarity), and the error as:

$$\mathcal{E} = \left( \frac{1}{50} \sum_{n=1}^{50} \frac{1}{HWT} (V_n^{'} - V_n)^2 \right)^{\frac{1}{2}} \tag{6}$$

where $V_n^{'}$ denotes the video generated using the implicit function $f_\theta$. Single-INR was optimized for 750 steps using Eqn. 1 taking $\sim 5.56$ minutes for each video ($\sim 4.63$ hours for 50 videos). Table. 1 presents quantitative

---

[1]The codebase, dataset, and pretrained models can be found at https://skymanaditya1.github.io/INRV

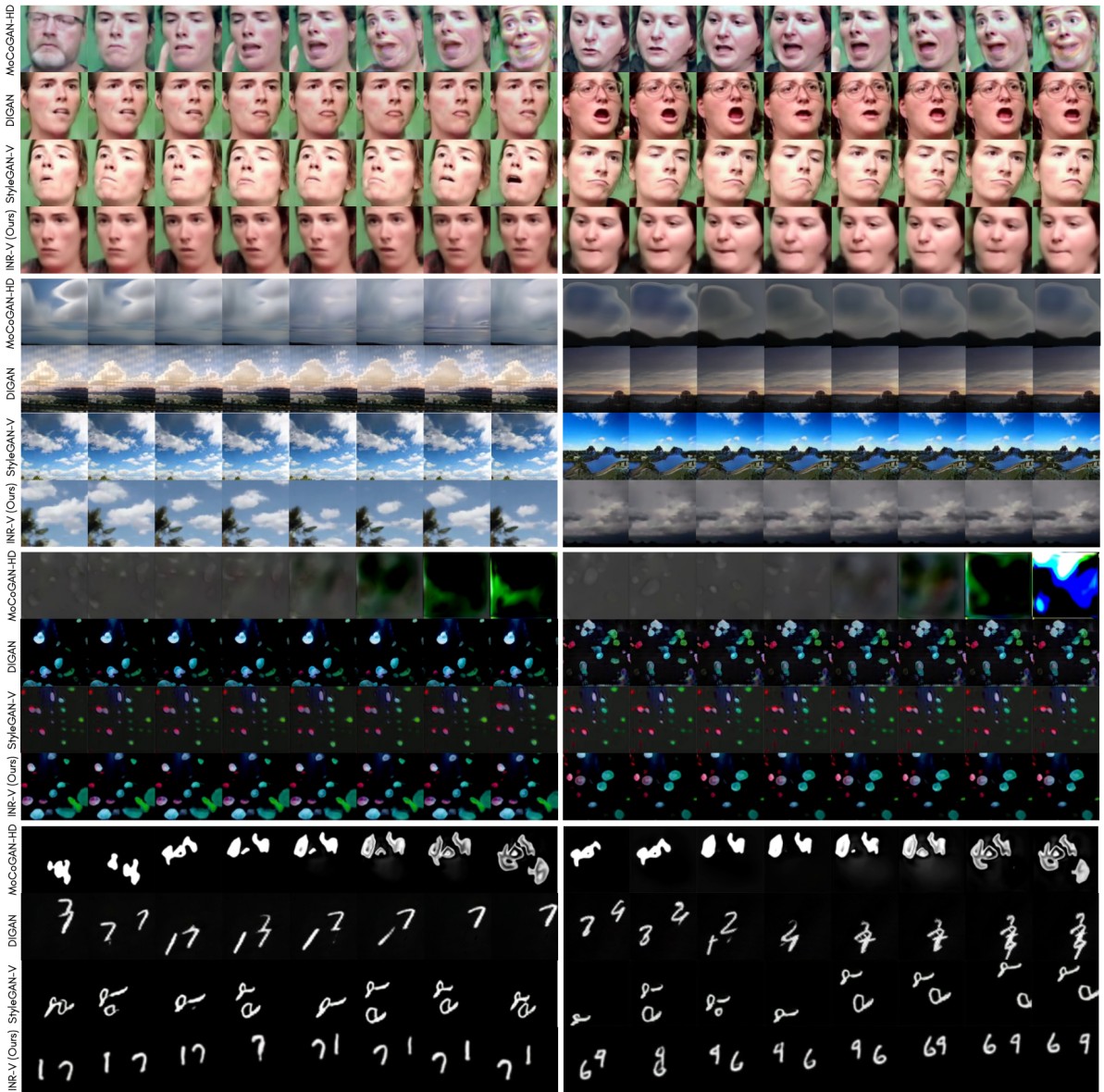

Figure 5: Examples of random videos generated on, from top to bottom, How2Sign-Faces Duarte et al. (2020), SkyTimelapse Xiong et al. (2017), RainbowJelly, and Moving-MNIST Srivastava et al. (2015). For Moving-MNIST, every $2^{nd}$ frame of 20 frames long videos and for other datasets, every $3^{rd}$ frame of 25 frames long generated are shown. Moving-MNIST and How2Sign-Faces are passed through VQVAE2 denoising network as described in Section 4.2

metrics on the videos reconstructed using Single-INR and INR-V. $PSNR_{FULL}$ computes the PSNR on the entire training dataset, $PSNR_{50}$ computes the metric on the selected 50 videos for comparison. As can be seen, although hypernetwork $d_{\Omega}$ is trained on huge datasets, it performs comparably with Single-INR. For RainbowJelly, it even outperforms Single-INR in SSIM metric and performs at par on SkyTimelapse. This indicates that $d_{\Omega}$ has learned to accurately generate INRs for complex spatio-temporal signals. Thus, INR-V can be used as a compression technique to compress 1000s of videos with minimal loss in perceptual quality.

## 4.2 Comparing INR-V with SOTA video generation networks

**Overview:** Fig. 5 and Table 2 present qualitative and quantitative comparisons respectively between MocoGAN-HD Tian et al. (2021), DIGAN Yu et al. (2022), StyleGAN-V Skorokhodov et al. (2021), and

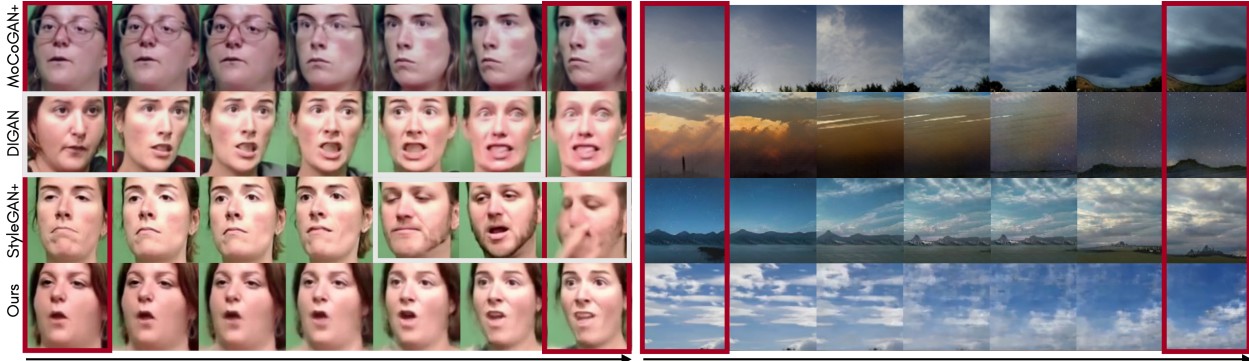

Figure 6: **Each cell displays $12^{\text{th}}$ frame of $25$ frames long generated videos**. The videos demonstrate interpolation between two given videos (in red boxes) by traversing along a trajectory in the latent space connecting the latent points of the given videos. Here, MoCoGAN+ and StyleGAN+ denote MoCoGAN-HD and StyleGAN-V. White boxes indicate a sudden transition in content (e.g. identity) or motion (e.g. pose).

INR-V. All models were trained from scratch. As we train the models on smaller datasets of $\sim 10{,}000$ datapoints, MoCoGAN-HD is trained on StyleGAN2-ADA Karras et al. (2020) image-generator backend. For each model, the best-performing checkpoint is selected for comparison.

**Evaluation:** As can be seen in Fig. 5, INR-V generates novel videos with coherent content and motion. MoCoGAN-HD fails to maintain the identity in a single video instance. For quantitative evaluation, we use the Frechet Video Distance (FVD) metric as implemented by StyleGAN-V. $\text{FVD}_{16}$ is computed on 2048 videos of 16 frames sampled at a resolution of $128 \times 128$. As can be seen in Table 2, INR-V outperforms the existing networks on Moving-MNIST and RainbowJelly and performs comparably on the remaining datasets.

**Enhancing INR-V's Visual Quality** Enhancing image and video quality has been an area of extensive research Yang et al. (2021); Chu et al. (2020); Liang et al. (2022); Chadha et al. (2020) with many breakthroughs. We propose that video generation can be partitioned into two stages (1) generating coherent content and motion (2) enhancing the visual quality. Note that, in the current work, our effort has been (1) to propose a novel continuous representation space for videos. We demonstrate (2) by developing a simple denoising network using a standard VQVAE2 Razavi et al. (2019). We train VQVAE2 as a frame-by-frame denoising autoencoder making one minor change: Instead of reconstructing the given low-quality input, we use the high-quality frame for computing the error. The low-quality inputs are the intermediate video instances reconstructed by INR-V during training. We train denoising VQVAE2 on How2Sign-Faces and Moving-MNIST. Appendix Fig. 18 demonstrates the results of the denoising network on blurry instances generated by INR-V. As can be seen from the quantitative metrics in Table. 2, using an additional denoising network improves the network's performance by $\sim 2\times$.

## 5 Applications of the continuous video space learned by INR-V

INR-V learns a continuous latent representation for videos allowing complex spatio-temporal video signals to be represented using a single latent point. In this section, we showcase the advantage of such a latent space through several demonstrated properties and comparisons. We also benchmark several tasks based on the inversion of 256 videos on How2Sign-Faces using full and incomplete video context.

### 5.1 Video Interpolation

Given two videos $V_1$ and $V_2$, a continuous video space should be able to make a gradual transition between the two videos such that every point along the trajectory between the two (1) produces a meaningful video and (2) shares content and motion properties from $V_1$ and $V_2$. We demonstrate this property in Fig. 1 and Fig. 6 with

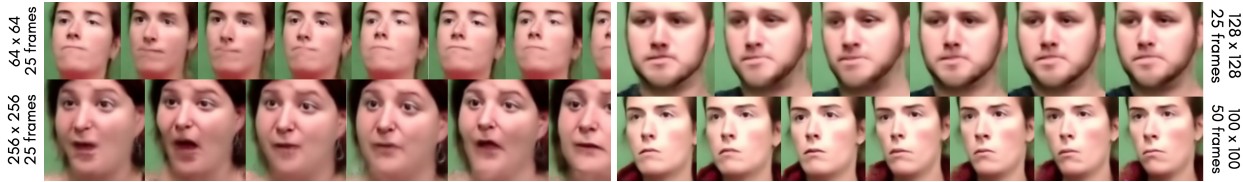

Figure 7: **INR-V direct inference on multiple resolutions and frame length**. INR-V trained on only 25 frames long $100 \times 100$ videos. Novel videos of multiple resolutions ($64 \times 64$, $128 \times 128$, $256 \times 256$) and video length (50) are directly generated on the trained model without any architectural change or finetuning. The images are not upto scale, please refer Appendix Fig. 25 for scaled representation.

Spherical Linear Interpolation (Slerp)[2]. Each cell in Fig. 6 demonstrate the $12^{th}$ frame of the 25 frames long videos. As can be seen, INR-V observed a gradual change in motion (pose, mouth movements, expressions, cloud shift) and content (identity, visibility of sun). The interpolated videos are spatio-temporally coherent (best seen in the supplementary video). Appendix Fig. 23 and Fig. 24 demonstrate the spatio-temporal transition on How2Sign-Faces and SkyTimelapse. As we represent an entire video in a single point in the continuous video space, interpolation is a natural operation that can be performed with INR-V.

**Comparisons:** Existing models have different motion and content codes; thus, to interpolate videos, intermediate content codes were interpolated between two videos by Slerp interpolation. INR-V does not have separate motion and content vectors; thus, videos can be interpolated directly using given video's latent points. As shown in Fig. 6, INR-V has a gradual transition in motion and content. For How2Sign-Faces, StyleGAN-V abruptly changes motion (cell 5-7), and DIGAN abruptly switches identity (cell 1-2, cell 5-6). This effect is

| MoCoGAN-HD | DIGAN | StyleGAN-V |
|---|---|---|
| 100.00 | 89.43 | 95.24 |

Table 3: Video interpolation user study: % of times INR-V interpolation was preferred over existing models.

highlighted in white boxes. This is expected as both of these architectures operate in the image space, and thus a gradual spatio-temporal transition is harder to achieve. We performed a user study on 30 users to qualitatively evaluate the interpolation quality of INR-V against the SOTA models and report the metrics in Table. 3. INR-V interpolation was randomly shown against either of the other three models. The users provided their preference on which interpolation looked smoother in terms of transition in content and motion. INR-V was preferred at least 85% more than all the SOTA networks. This demonstrates the continuous nature of the video space learned by INR-V.

## 5.2 Multi-Resolution and Multi-Length Inference

In Fig. 7 we show INR-V trained on videos of only $100 \times 100$ resolution with 25 frames per video, generating novel videos of multiple resolutions and lengths, maintaining the content and motion quality of the output. An underlying property of INRs is a continuous representation of the signal given as $f_\theta(h, w, t) \rightarrow$ RGB. This enables the model to understand a continuous property of the signal making it agnostic to the dimension. We show quantitative metrics on INR-V inferred at multiple resolutions and compare INR-V with existing SOTA superresolution techniques Chen et al. (2022) in Appendix A.5.

## 5.3 Video Inversion and its applications

Inversion has been widely adopted in many applications prominently for images. StyleGAN2 Karras et al. (2019) is extensively used for image inversion enabling many downstream image editing tasks such as changing the emotion, age, or gender of a given face. In video inversion, we aim to invert a given video back into the latent space of a pretrained video generation network. Existing methods perform frame-by-frame inversion to individually invert the context code for each frame and the motion code for the video. In INR-V, we only

---
[2] https://splines.readthedocs.io/en/latest/rotation/slerp.html

need to invert to a single latent code that can be achieved through a simple optimization objective:

$$\operatorname*{argmin}_{m_z} \frac{1}{T}\frac{1}{W}\frac{1}{H}\sum_{T}^{t=1}\sum_{W}^{w=1}\sum_{H}^{h=1}(f_{\theta_z}(h,w,t) - RGB_s)^2 \quad \text{where} \quad \theta_z = h_\Omega(m_z) \tag{7}$$

where $m_z$ is the latent point for a video instance $V_z$. Fig 8 shows the qualitative demonstration of INR-V inversion trained on How2Sign-Faces for a video outside of the training dataset $\mathcal{D}$.

**Video Completion:** Key categories of 'video synthesis' include future frames prediction (future prediction), completing the video between frames (frame interpolation), and predicting the missing part of the video (video inpainting). In INR-V, a video $V_z$, represented by a single latent code $m_z$ can be generated without any additional knowledge. Thus, all the above operations can be performed using a modified optimization operation based on Eqn 7 on the seen part of the video given as:

$$\operatorname*{argmin}_{m_z} \frac{1}{S}(f_{\theta_z}(h_s, w_s, t_s) - RGB_s)^2$$
$$\text{and} \quad \theta_z = h_\Omega(m_z) \tag{8}$$

where $S$ is the number of context points, $h_s$, $w_s$, and $t_s$ are the context points of $V_z$ seen at the time of optimization. With the optimized $m_z$, the full video can simply be generated back with INR-V. Fig. 8 demonstrates the results for the various operation on a video outside of $\mathcal{D}$ with $\sim 2.5$ minutes of optimization on a single 12 GB NVIDIA GTX 2080ti GPU. As can be seen, the network is able to regress to a latent corresponding to the given identity while preserving finer details like spectacles, mouth shape, pose, etc. In the case of 'Video Inpainting', the network understands the person's pose. For 'Frame Prediction', although the pose does not match the ground truth, the overall video is coherent. In 'Frame Interpolation', the model is able to generate a coherent context between two frames, including the pose, expressions, identity, and mouth movements. In 'Sparse Inpainting', we randomly set 25% of all the video pixels as the context points for optimization. Even with very sparse context, INR-V is able to regress to the correct specifications including the finer content details.

**Video Superresolution through inversion:** Video Superresolution is the task of enhancing the

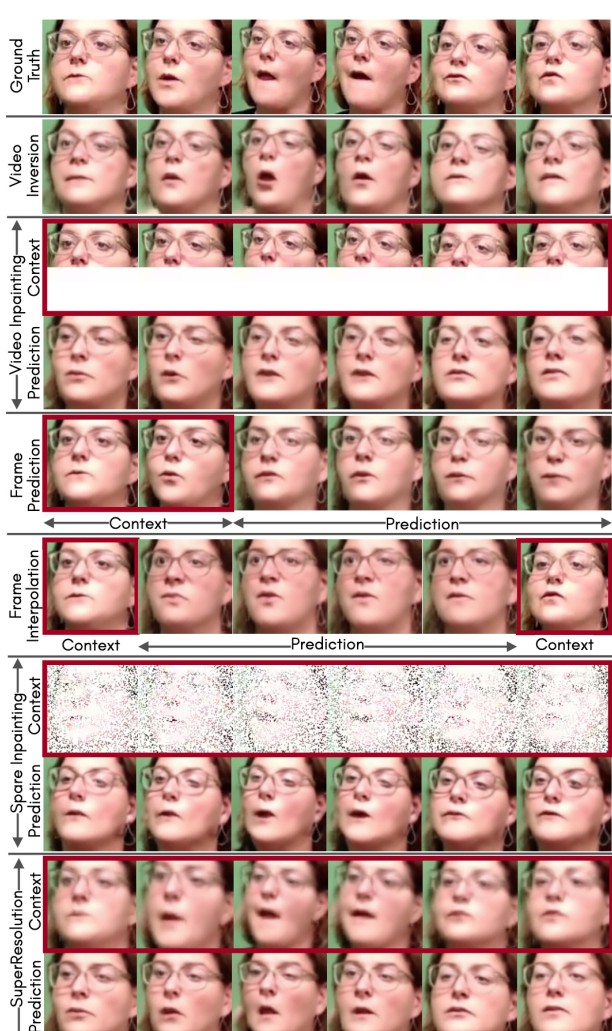

Figure 8: **Video Inversion and it's applications.** INR-V can be directly used for several tasks by simply inverting a video to its latent point based on the given context. We demonstrate some qualitative results.

resolution of a given video. Recent works such as Chu et al. (2020); Liang et al. (2022); Sajjadi et al. (2018); Chadha et al. (2020); Wang et al. (2019); Chen et al. (2022) have made significant progress in video super-resolution, showcasing 4× enhancement. INR-V can directly superresolve seen video instances as showcased in Appendix Table. 7. For unseen instances, combining the capability of video inversion and multi-resolution video generation, INR-V can superresolve a given video $V_z$ of a lower resolution (say $32 \times 32$) simply as following: (1) Invert $V_z$ at the smaller resolution to obtain $m_z$. (2) Render $V_z$ from $m_z$ directly at a higher resolution (say $256 \times 256$). In Fig. 8, we demonstrate the qualitative results on a video outside the training dataset. The video was optimized at $32 \times 32$ for $\sim 2.5$ minutes. The inverted video was then superresolved at a scale factor of 8× to $256 \times 256$. Additional details are present in Appendix A.5.

**Quantitative Evaluation:** To quantify the performance of INR-V, we prepare a comparison set by randomly sampling 256 videos outside of the training set. We compare against DIGAN on the tasks of Video Inversion, Video Inpainting, Frame Prediction, Frame Interpolation, and Sparse Interpolation and against StyleGAN-V on the task of Video Inversion. Since DIGAN is based on INRs, it can invert incomplete frames, however, StyleGAN-V expects a full frame for backpropagation. Thus we do not compare

| Task | Method | GT-ID ↑ | TL-ID ↑ | TG-ID ↑ | Context-L1 ↓ | PSNR ↑ | SSIM ↑ | Cost ↓ |
|------|--------|---------|---------|---------|--------------|--------|--------|--------|
| Inv. | DIGAN | 0.652 | 0.953 | 0.9599 | 45.08 | 19.59 | 0.653 | ∼ 4.25 |
|      | Style-V | **0.804** | **0.985** | **0.998** | 42.16 | 19.65 | 0.665 | ∼ 3.25 |
|      | INR-V | 0.770 | 0.950 | 0.950 | **5.25** | **21.21** | **0.773** | ∼ **2.75** |
| Inp. | DIGAN | 0.628 | **0.960** | **0.969** | 45.80 | - | - | ∼ 4.25 |
|      | INR-V | **0.758** | 0.948 | 0.939 | **4.83** | - | - | ∼ **2.75** |
| Pre. | DIGAN | 0.603 | 0.940 | 0.928 | 40.26 | - | - | ∼ 4.25 |
|      | INR-V | **0.703** | **0.946** | **0.932** | **4.72** | - | - | ∼ **2.75** |
| Int. | DIGAN | 0.653 | 0.925 | **0.921** | 48.66 | - | - | ∼ 4.25 |
|      | INR-V | **0.702** | **0.928** | 0.905 | **7.46** | - | - | ∼ **2.75** |
| Spr. | DIGAN | 0.718 | 0.961 | 0.967 | 46.24 | 19.74 | 0.671 | ∼ 4.25 |
|      | INR-V | **0.768** | **0.968** | **0.974** | **5.29** | **22.35** | **0.774** | ∼ **2.75** |
| Sup. 4× | Bicubic | 0.808 | 0.923 | 0.903 | - | 28.36 | 0.906 | - |
|      | VideoINR | **0.939** | **0.982** | **0.974** | - | **32.86** | **0.957** | - |
|      | INR-V | 0.734 | 0.911 | 0.903 | 4.92 | 21.94 | 0.742 | ∼ **2.75** |

Table 4: Comparison of INR-V on various video inversion tasks: Video Inversion (Inv.), Video Inpainting (Inp.), Frame Prediction (Pre.), Frame Interpolation (Int.), Sparse Interpolation (Spr.), and Superresolution (Sup.). Comparison set is made of 256 videos outside of the training dataset. Metrics used for evaluation is explained in Sec. 5.3. Cost denotes the time to optimize a single video instance in minutes.

with StyleGAN-V on the other tasks. For the task of Superresolution, we compare against Bicubic Upsampling and VideoINR at a scale factor of 4× from $32 \times 32$ to $128 \times 128$.

We evaluate on the following metrics: (1) PSNR, (2) SSIM, (3) Temporally Locally (TL-ID) and Temporally Globally (TG-ID) Identity Preservation, (4) Context-L1, and (5) Ground Truth Identity (GT-ID) Match. TL-ID and TG-ID were proposed in Tzaban et al. (2022). They evaluate a video's identity consistency at a local and global level. For both metrics, a score of 1 would indicate that the method successfully maintains the identity consistency of the original video. Context-L1 computes the L1 error on the inverted videos at the given context points. An error of 0 would indicate that the inversion is perfect. GT-ID measures the match in identity between the ground truth and the inverted video. DeepFace[3] face features are extracted for both the videos, and the cosine similarity is computed between the extracted features. Since there is no single correct prediction for tasks like 'Future Frame Prediction', 'Frame Interpolation', and 'Video Inpainting', we do not evaluate these tasks on PSNR and SSIM.

As can be seen, INR-V outperforms all the existing networks in most of the metrics on video inversion and the proposed inversion tasks, except 'Superresolution', indicating the advantage and robustness of the proposed space. For the task of Superresolution, INR-V performs comparably with Bicubic and VideoINR. However, unlike these works that directly superresolve a video, INR-V first inverts the low resolution video to generate a high resolution video. Such a mechanism opens several possibilities, such as inverting a low resolution incomplete video (missing frames due to corruption) to a high resolution video with full context.

## 6 Conclusion

We present INR-V, a continuous video representation network. Unlike existing architectures that extend superior image generation networks for generating videos one frame at a time, we use implicit neural representations to parameterize videos as complete signals allowing a meta-network to encode it to a single latent point. Given enough examples, the meta-network learns a continuous video space as demonstrated through video interpolation and inversion tasks. INR-V generates diverse coherent videos outperforming many existing video generation networks. INR-V opens the door to a multitude of video-based tasks and removes the dependency on an image generator. To showcase this, we propose several downstream tasks and observe that INR-V outperforms the existing works on a majority of these tasks. This demonstrates the advantages and potential of a continuous video space and we hope to encourage research in this direction.

---

[3] https://github.com/serengil/deepface

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

# A  Appendix

## A.1  Ablation

### A.1.1  Effect of Regularization

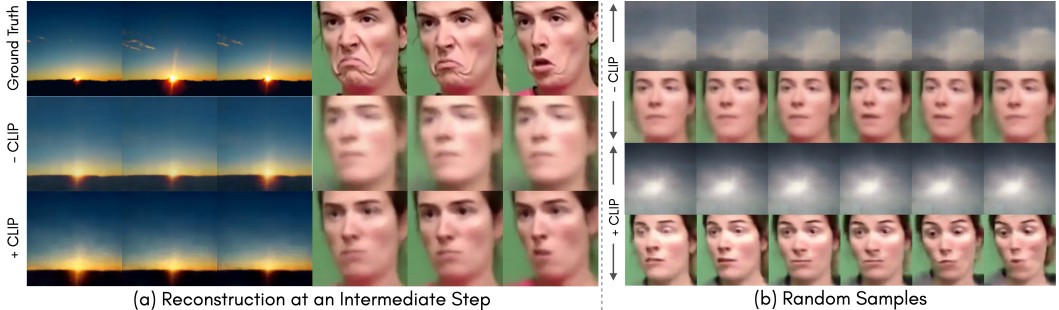

(a) Reconstruction at an Intermediate Step        (b) Random Samples

Figure 9: Qualitative results of CLIP regularization. Intermediate results are shown after 30 hours of training on 2 NVIDIA GTX 2080ti GPUs. (a) Video reconstruction quality. CLIP regularization enables the meta-network to model the INRs with finer details. (b) Videos generated by random sampling. CLIP regularization improves the quality of the sampled videos and encourages variation in the implicit representations.

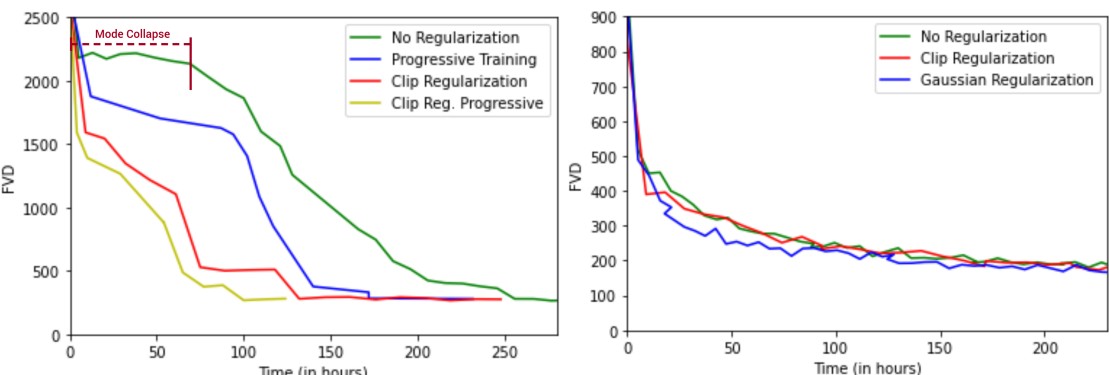

Figure 10: Convergence rate for different regularization schemes on RainbowJelly (left) and SkyTimelapse (right) datasets. RainbowJelly consists of $\sim 34$K datapoints and SkyTimelapse consists of $\sim 2$K datapoints. As can be observed, SkyTimelapse being a relatively smaller dataset performs equally well on all the regularization schemes. RainbowJelly performs worst without any regularization facing mode-collapse for about the first $\sim 75$ hours. Progressive training and CLIP regularization help INR-V converge the fastest.

In this section, we compare the training time and the performance of INR-V (1) with/without CLIP regularization and (2) with/without progressive training. Fig. 9 presents the qualitative results on INR-V after 30 hours of training on 2 NVIDIA GTX 2080ti GPUs. Fig. 10 plots the rate of convergence on RainbowJelly (left) and SkyTimelapse (right) datasets on the same training setup. We also show Gaussian regularization with INR-V by adding the following additional loss term to the overall loss term in Eqn. 2:

$$\delta D_{KL}(\, \mathcal{N}(\mu, \sigma) \, || \, \mathcal{N}(0, 1) \,) \tag{9}$$

where $\mu$ and $\sigma$ denote the mean and standard deviation over the latent codes $\{m_n\}_{n=1}^{N}$ and $\delta$ is a hyperparameter. In our experiments, $\delta = 1.0$.

As can be observed from Fig. 9, reconstruction quality is much worse without CLIP. This is expected as Video-CLIP (see Fig. 4) assigns semantic meaning to the initialized codes for each video instance. As we observe the

novel video instances generated using this model (Fig. 9, right), we already see a motion emerging with expressive faces. As can be observed from Fig. 10, on RainbowJelly dataset (made of 34526 instances), INR-V takes more than 250 hours ($\sim$ 11 days) to converge without any regularization scheme or progressive training. With progressive training, the convergence time is drastically reduced to less than 180 hours ($\sim$ 7.5 days). The best performance is achieved when progressive training is done along with CLIP regularization where the convergence occurs in less than 120 hours ($\sim$ 5 days) on 2 NVIDIA GTX 2080ti GPUs. On SkyTimelapse dataset (made of 1803 instances), INR-V converges equally on either of the regularization schemes. With a Gaussian prior, we observed a slight advantage in terms of convergence time. Fig. 11 plots an additional comparison on RainbowJelly with progressive training on three different regularization methods: Gaussian, CLIP, and no regularization.

The graphs indicate that CLIP regularization is more suitable for a larger dataset like RainbowJelly, however for a smaller dataset like SkyTimelapse (Fig 10, right), Gaussian regularization is more effective. INR-V performs equally well on either of the training schemes, given enough time to train. This indicates that the generation capabilities is inherent to the proposed architecture, whereas the different training schemes help in stabilizing the training and thus, lead to a faster convergence. Additional insights are provided in Appendix A.4.

### A.1.2 Effect of the size of the codebook

Fig 12 presents a comparison between the FVDs of INR-V when trained on varying number of video instances on the RainbowJelly dataset. FVD is computed against the entire dataset made of $34,526$ samples As can be seen from the graph, the performance of INR-V deteriorates as the number of video instances reduce. However, the effect of the added instances is marginal as the codebook size increases; the FVD improving by only 12% when going from 10K to 34K (24K additional instances) video instances. However, the FVD improves by 20% when the codebook size increases from 500 to 1000 (500 additional instances) video instances.

### A.1.3 Progressive Training with different Initializations

To stabilize the hypernetwork training, we train our network progressively as explained in Sec. 3.3. In this section, we compare the performance of INR-V on training with different video intializations. We randomly assign videos for each stage of the training and the random assignments differ across the different initializations. As shown in Fig. 3.3, INR-V takes about the same time to converge for all of them.

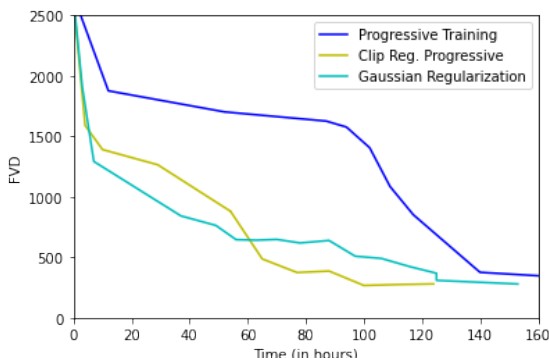

Figure 11: Convergence rate for different regularization schemes on RainbowJelly dataset made of $\sim$ 34K instances. All the models are trained progressively. CLIP Regularization leads to the fastest convergence.

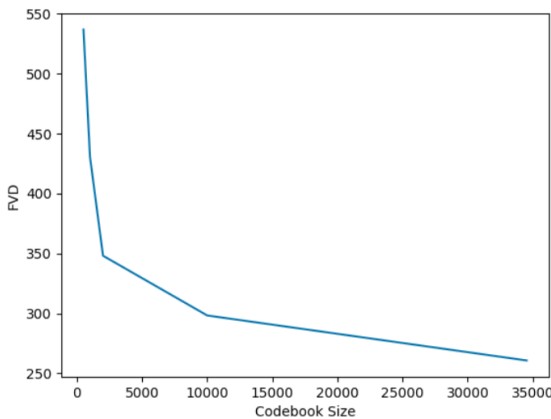

Figure 12: Performance of INR-V when trained on varying number of video instances (codebook size). FVD is computed against the full dataset.

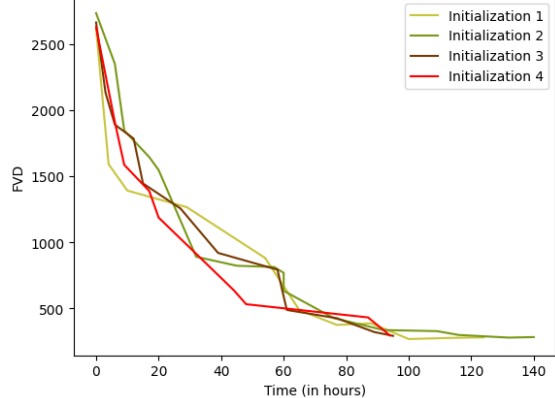

Figure 13: INR-V on progressive training with different initializations: the order of video selection for each stage differ across the initialization.

## A.2   INR-V Implementation Details

The implicit neural representation $f_\theta$ is an MLP with three 256-dimensional hidden layers. Each hidden layer is passed through ReLU activations. The hypernetwork $d_\Omega$ is a set of MLPs. Each MLP predicts the weights for a single hidden layer and the output layer of $f_\theta$. Each MLP has three 256-dimensional hidden layers. CLIP embeddings are 512-dimensional vectors, Video-CLIP encodes the CLIP embeddings of each frame through three 512 dimensional, GRU layers. As shown in Fig. 4, Video-CLIP produces 512-dimensional video-level embedding $g_n$. $c_n$ is a 512-dimensional context vector that is regressed in an auto-decoding fashion during training. $\phi$ is made of 3-hidden layers that takes a 1024-dimensional vector as input (concatenation of $g_n$ and $c_n$) and produces $m_n$, a 128-dimensional instance code of $V_n$, as the input for $d_\Omega$. The input to $f_\theta$ is a periodic positional encoding of $(\{h\}_{h=1}^H, \{w\}_{w=1}^W, \{t\}_{t=1}^T)$ as implemented in Sitzmann et al. (2021). Adam optimizer is used with a learning rate of $1e-4$ during training and $1e-2$ during inversion tasks. No scheduler is used. Progressive training is done at a power of 10 where $i^{\text{th}}$ stage is made of $\min(10^i, N)$ examples. $i = 0 \dots \mathcal{K}$ such that $10^{\mathcal{K}+1} < N+1$, where $N$ is the total number of training samples. Each stage except the last stage is trained until the reconstruction error reaches a threshold of $1e-3$.

## A.3   Comparison of Computational Complexity of INR-V against 3D Convolutional models

We compare the computational complexity of INR-V against standard implementations of 3D convolution-based video generation models of varying spatial and temporal dimensions. We call these models as "3DConv". 3DConvs do not generate any meaningful output and are used solely to compare the computation costs against INR-V. They comprise of several transpose 3D convolution-based upsampling layers and take a fixed 128-dimensional latent vector as input. For comparison against INR-V, we gradually vary their spatial dimension from $128 \times 128$ to $1024 \times 1024$ by keeping the temporal dimension fixed to 25

|  | $128 \times 128$ | $256 \times 256$ | $512 \times 512$ | $1024 \times 1024$ |
|---|---|---|---|---|
| INR-V | 1.68 | 6.71 | 26.84 | 107.36 |
| 3DConv | 252.71 | 691.76 | 3860.77 | OOM |
|  | 25 frames | 50 frames | 75 frames | 100 frames |
| INR-V | 6.71 | 13.42 | 20.13 | 26.84 |
| 3DConv | 691.76 | 2036.05 | 3719.98 | 6673.27 |

Table 5: Comparison of computational complexity of INR-V against 3D convolutional-based video generative models for different spatial and temporal dimensions. The computational complexity is the total number of multiply-add Giga operations denoted by GMAC (Multiply-Add Cumulation).

frames. To generate a video of resolution $128 \times 128$, three 3D convolution-based upsampling layers are used. An additional upsampling layer is added for every subsequent jump in the spatial dimension. Next, we vary the temporal dimension for 25, 50, 75, and 100 frames by keeping the spatial extent and the number of layers fixed to $256 \times 256$ and four, respectively, and adjusting the kernel size corresponding to the temporal dimension. The stride, kernel size, and padding are appropriately adjusted for all the models. The batch size for comparison is fixed to 1 for both 3DConv and INR-V. As can be seen in Table. 5, the number of operations (MAC) increases drastically as the spatial dimension increases for the 3DConvs. It becomes prohibitively expensive to generate videos of higher spatial dimensions. For example, generating a single video of 25 frames of dimension $1024 \times 1024$ results in out of memory (OOM) on a single NVIDIA GTX 2080 ti GPU with a memory of 12 GB. In summary, MAC remains hundreds of orders of magnitude lower for INR-V compared to its 3D convolution-based counterparts as it mainly comprises inexpensive MLPs.

## A.4   Insights on the learned latent space

Unlike the existing video generation networks that are conformed to a predefined latent space (Gaussian or Uniform), INR-V learns a space that best fits a given distribution. The experiments (Sec. 5.1 and Sec. 5.3) and our observations indicate that the learned space is continuous, supports inversion, and smooth video interpolations. Thus, such a space learns a structure in the dataset. For instance, we observe a smooth transition across different poses, expressions, mouth movements, and identity on How2Sign-Faces (best viewed in the Supplementary Video). Therefore, novel instances are observed as we traverse the path between seen latent points A and B. We use this property to generate novel videos by sampling latent points

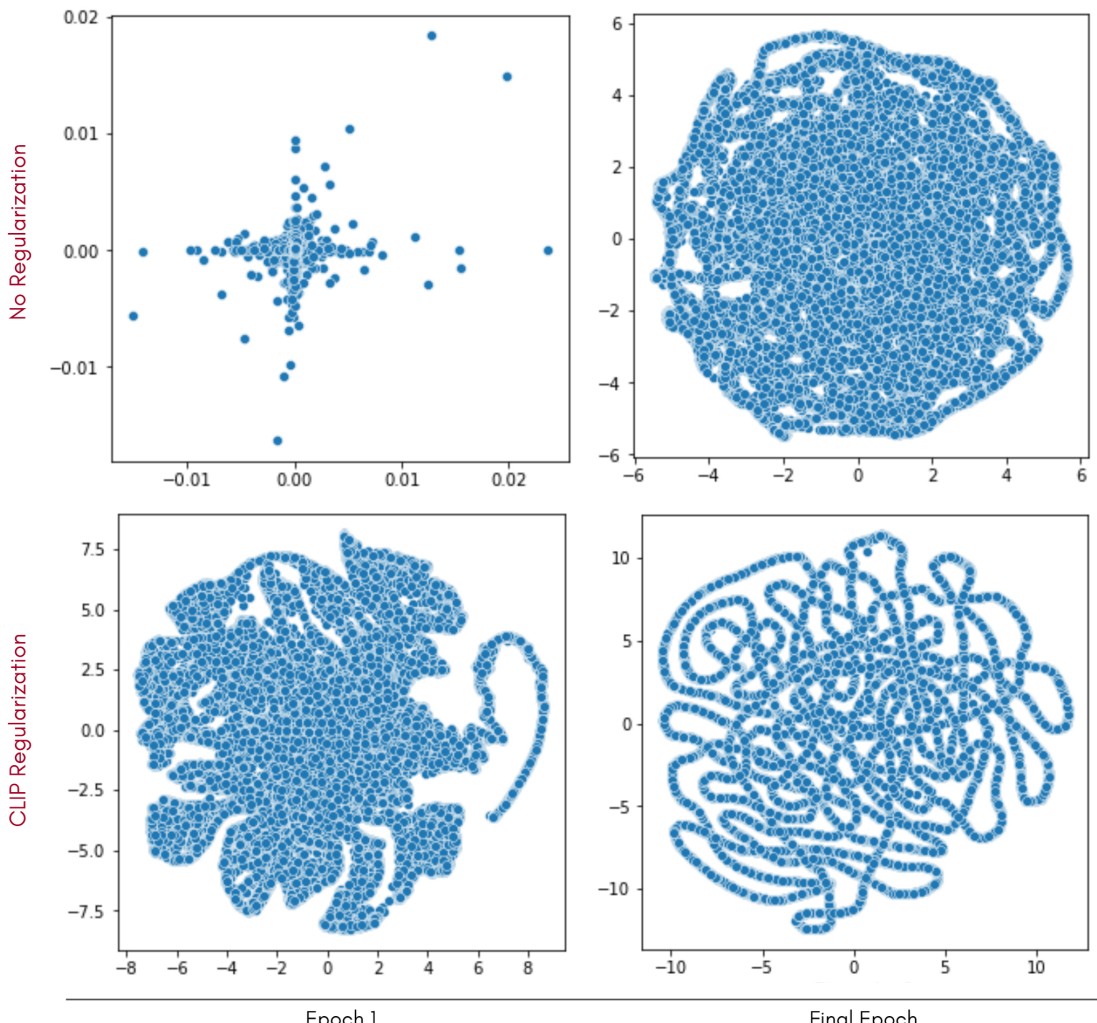

Figure 14: t-SNE visualization of the latent codes ($m_n$) learned by INR-V with and without CLIP regularization on RainbowJelly. In Epoch 1, the latent codes are bounded within a very small region when INR-V is trained without any regularization (top-left). With CLIP regularization, the latents are more spread out (bottom-left). In the final epochs (top & bottom right), both latent representations have spread out farther. INR-V without regularization forms a denser space; and with regularization converges faster. Both spaces are continuous, form meaningful interpolations, and support inversion.

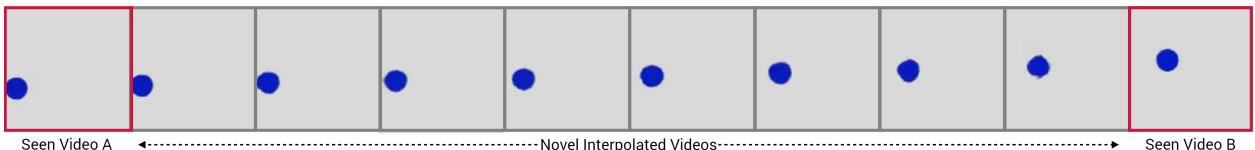

Figure 15: We demonstrate the ability of INR-V to learn the underlying structure of a dataset. To do so, we create a toy video dataset called BouncingBall consisting of a blue ball bouncing horizontally at different heights. Given 50 instances of such videos, INR-V learns the structure and we demonstrate the learned structure through interpolation. In this example, we show the 16$^{\text{th}}$ frame of each video with the novel interpolated videos in gray boxes. The videos in red boxes are seen during training. We can observe a correct interpolation with smooth transition in motion: the height and the horizontal position of the ball gradually shifts from bottom to top and left to right respectively.

through Slerp interpolation. In this section, we aim to validate the space learned by INR-V and answer the following question: what does INR-V learn? We analyse this in two ways: (1) by visualizing the latent codes learned by INR-V through t-SNE visualization and (2) by training INR-V on a structured toy dataset to see if it learns the underlying structure.

### A.4.1 t-SNE visualization

Fig. 14 plots t-SNE[4] on the latent codes learned by INR-V on the RainbowJelly dataset. We see a clear pattern of "interpolation" occurring in the learned latent space in both cases (with and without CLIP regularization). No progressive training is used in this visualization. At the end of the first epoch, the latent codes are tightly bounded when trained without CLIP regularization. As the model is trained with CLIP, the latent vectors are more spread out possibly in a semantically meaningful manner. In both the cases, by the final epoch, we observe patterns of interpolation evolve.

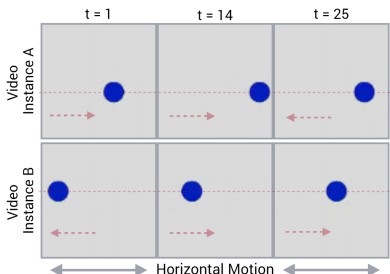

Figure 16: BouncingBall dataset with an infused structure. Each video instance is $100 \times 100$ and has a ball bouncing horizontally at a specific height. Red lines are added to show the height of the ball and is not a part of the videos.

### A.4.2 Learning Bouncing Ball

In this section, we want to analyze if INR-V can learn the structure of a dataset. To do so, we generate a toy dataset, BouncingBall, with an artificially infused structure. The dataset is made of 50 video instances where each video instance consists of a blue ball bouncing horizontally at different heights as shown in Fig. 16. The videos are of dimension $100 \times 100$ and are 25 seconds long each.

As shown in Fig. 15, INR-V is able to learn the infused structure when trained on BouncingBall. We demonstrate interpolated videos by sampling intermediate points through Slerp interpolation. The intermediate videos (grey boxes) demonstrate a smooth interpolation, and have heights and horizontal displacements gradually varying from Video A to B (red boxes). An example of an intermediate video shown in Fig. 17.

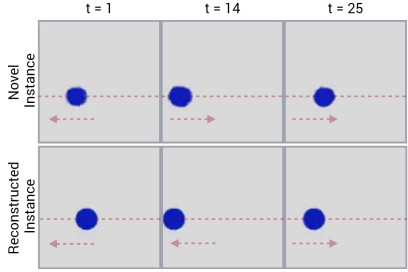

### A.5 Inferring at Multiple Resolutions and Multiple Lengths

An underlying property of INRs is a continuous representation of the signal. This allows inferring INR-V on multiple spatial and temporal resolutions directly without changing the model's architecture or additional finetuning. To generate a video of an arbitrary dimension $\hat{H} \times \hat{W} \times \hat{T}$, those many number of equally spaced points are sampled between $[-1, 1]$. In Table 6, we report $\text{FVD}_{16}$ scores on random videos

Figure 17: Videos generated by INR-V on BouncingBall. Novel video is generated at an unseen height. Red lines are for demonstration and not generated.

generated on INR-V with varying spatial dimensions on 25 frames. To infer at multiple resolutions, INR-V pretrained on $100 \times 100$ dimensional videos of 25 frames was used. Note that the FVD scores do not degrade even at higher spatial resolutions. Additionally, we compare INR-V with existing SOTA superresolution techniques Chen et al. (2022) in Table 7 on 2048 videos randomly sampled from the RainbowJelly dataset. As can be seen, INR-V performs comparably with methods on the task of superresolution. Moreover, INR-V can be superresolved to any arbitrary resolution ($3.6\times$) and aspect ratio. Please note that we do not solve the task of superresolution but rather show superresolution as a potential application of our work.

### A.6 Discussion

**Limitations.** Although INR-V has learned a powerful video space demonstrating several intriguing properties, the videos generated by the model are sometimes blurry. This is prominent when moving away to unseen points in the video space far from the seen instances. Fig. 18 demonstrates the enhancement on one

---

[4] https://lvdmaaten.github.io/tsne/

such blurry sample. This is done by training a standard VQVAE2 network in a denoising fashion (please refer to Sec. 4.2). However, the entire process is broken into generating a relatively lower quality output and relying on a second network to improve its quality. A single end-to-end network capable of retaining the demonstrated powerful properties while generating high-quality videos is a potential future work.

Another limitation of INR-V is generating infinitely long videos. Although coupling the content and time into a single latent has clear advantages, it removes the network's ability to leverage the temporal dimension separately and find infinitely long temporally coherent paths in the image space. This can be tackled by training INR-V to encode video segments of multiple lengths in a single space (1 to 50 or more frames long video segments). A temporally and semantically coherent trajectory between these video segments can then be learned. Such a generation technique would directly leverage video segments and potentially remove repetitions in the long videos. We believe that leveraging a video space for generating infinitely long videos at multiple resolutions presents an interesting and exciting direction for future research.

Lastly, we observed that INR-V does not learn a meaningful representation space when trained on datasets like UCF-101 that have extreme diversity and limited structure in motion. A similar issue was observed when training the baseline models on such datasets. However, INR-V can be trained on a single action class of UCF-101 (such as JumpRope) to learn a meaningful representation space even with significant camera motion and in-video subject movement. A single action class limits the visual and motion diversity in the dataset.

|  | $128 \times 128$ | $256 \times 256$ | $360 \times 360$ |
|---|---|---|---|
| INR-V (Ours) | 260.72 | **232.43** | 251.14 |

Table 6: $FVD_{16}$ ($\downarrow$) metrics on random video generation at multi-resolution on INR-V. Training was done on only $100 \times 100$ dimensional videos of 25 frames. Inference was taken directly on multiple resolutions without any finetuning or architectural changes.

|  | $2\times$ $200 \times 200$ | | $3\times$ $300 \times 300$ | | $3.6\times$ $360 \times 360$ | |
|---|---|---|---|---|---|---|
|  | PSNR $\uparrow$ | SSIM $\uparrow$ | PSNR $\uparrow$ | SSIM $\uparrow$ | PSNR $\uparrow$ | SSIM $\uparrow$ |
| Bicubic | 31.53 | 0.884 | 32.13 | **0.915** | 32.31 | **0.920** |
| VideoINR | **31.59** | 0.883 | **33.01** | 0.913 | - | - |
| INR-V | 28.62 | **0.892** | 29.17 | 0.894 | 29.05 | 0.896 |

Table 7: Quantitative metrics on video superresolution using INR-V and SOTA superresolution networks on video instance seen at the time of training. INR-V was trained at $100 \times 100$ video resolution. INR-V performs comparably with the SOTA superresolution networks.

**Broader Impact.** The potential negative impact of our work is similar to existing image-based and video-based GANs: creating "photorealistic-deepfakes" and using them for malicious purposes. Our simple training strategy makes it easier to train a model which produces realistic-looking videos. However, this is partly addressed for the following reasons: (1) Even though our network produces diverse novel videos, the perceptual quality of our generated videos falls short of the existing state-of-the-art image-based generators that produce high-resolution images. (2) The availability of high-

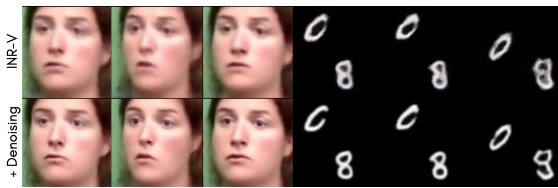

Figure 18: Denoising VQVAE2 reconstructions to enhance the visual quality of relatively blurry videos generated by INR-V. Please refer Sec. 4.2.

quality video datasets limits the intended malicious use of this codebase. Despite these limitations, we believe that the potential of our work far outweighs its limitations. A continuous video representation space offers tremendous applications in areas requiring video prediction, interpolation, and conditional video generation. E.g. pedestrian trajectory prediction is an important area of research for self-driving cars. Pedestrian trajectory prediction through future frame generation can serve to reduce accidents in fully-autonomous vehicles in the future. Similarly, conditional video generation can be used for synthesizing novel sign language videos that can be integrated into schools and universities to encourage and enable hard-of-hearing students.

## A.7 Additional Qualitative Results

We encourage our readers to view the supplementary video results of INR-V. Fig. 19 presents the real video instances in the training set. Fig. 20 and Fig. 21 presents qualitative results on the reconstruction of video instances from different training datasets. Fig 22 presents random videos generated by INR-V on different datasets. Fig 23 and Fig. 24 present spatio-temporal view of video interpolations on How2Sign-Faces and

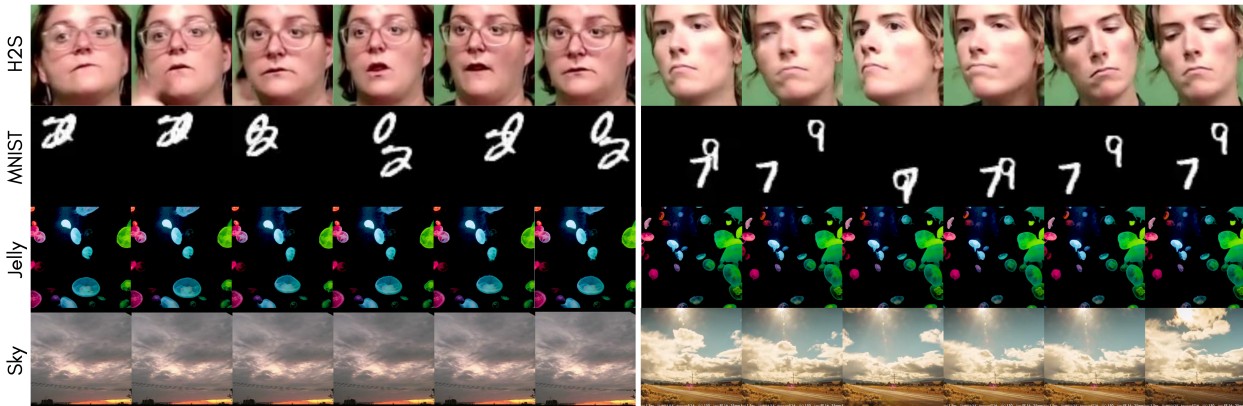

Figure 19: Examples of real videos instances of How2Sign-Faces (H2S) Duarte et al. (2020), Moving-MNIST (MNIST) Srivastava et al. (2015), RainbowJelly (Jelly), and SkyTimelapse (Sky) Xiong et al. (2017) datasets.

SkyTimelapse respectively. Fig. 25 presents the random generation of INR-V on multiple resolutions starting from $32 \times 32$ to $256 \times 256$ jumping a scale factor of $8\times$. The visualization is up to scale, and one can see the scale jump. INR-V can also be inferred at multiple frame rates. The supplementary videos include inferences at 50 frames. Fig. 26 - Fig. 32 present the qualitative results and comparisons on the proposed inversion tasks. Fig. 30 presents an example of multi-modal future segment prediction. Additional results on several inversion tasks can also be found in the supplementary video.

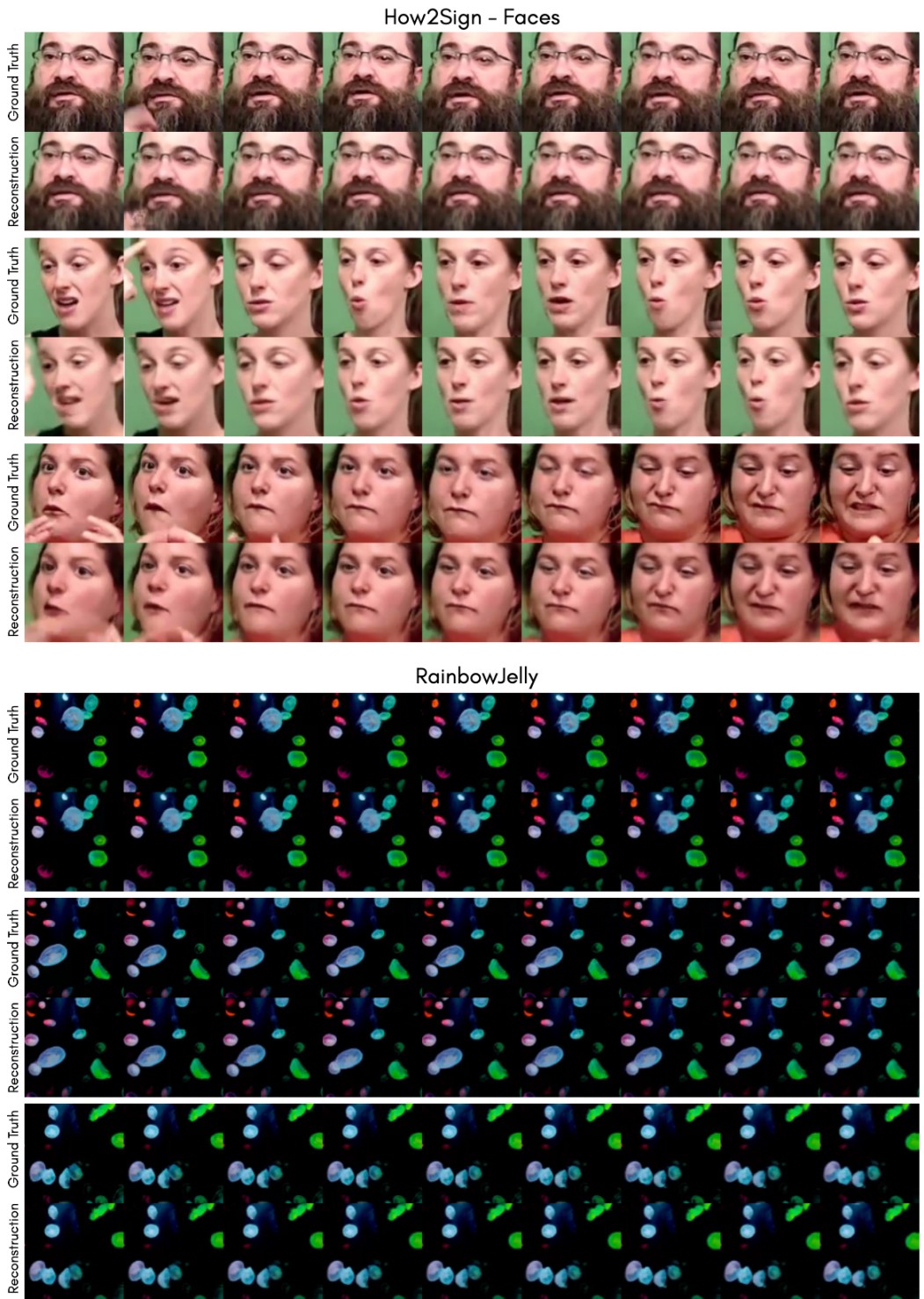

Figure 20: Examples of video instances in the training set reconstructed by INR-V.

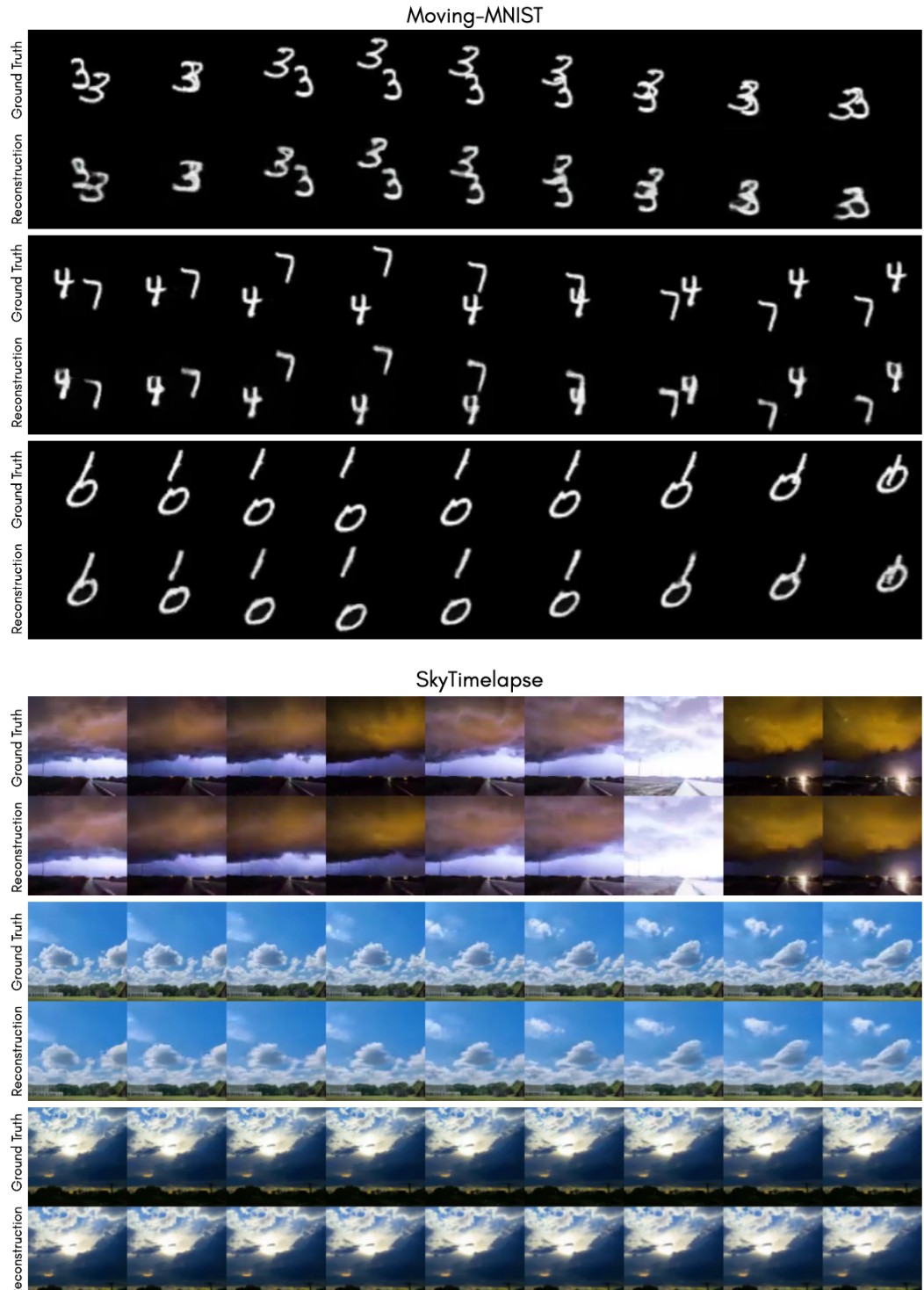

Figure 21: Examples of video instances in the training set reconstructed by INR-V.

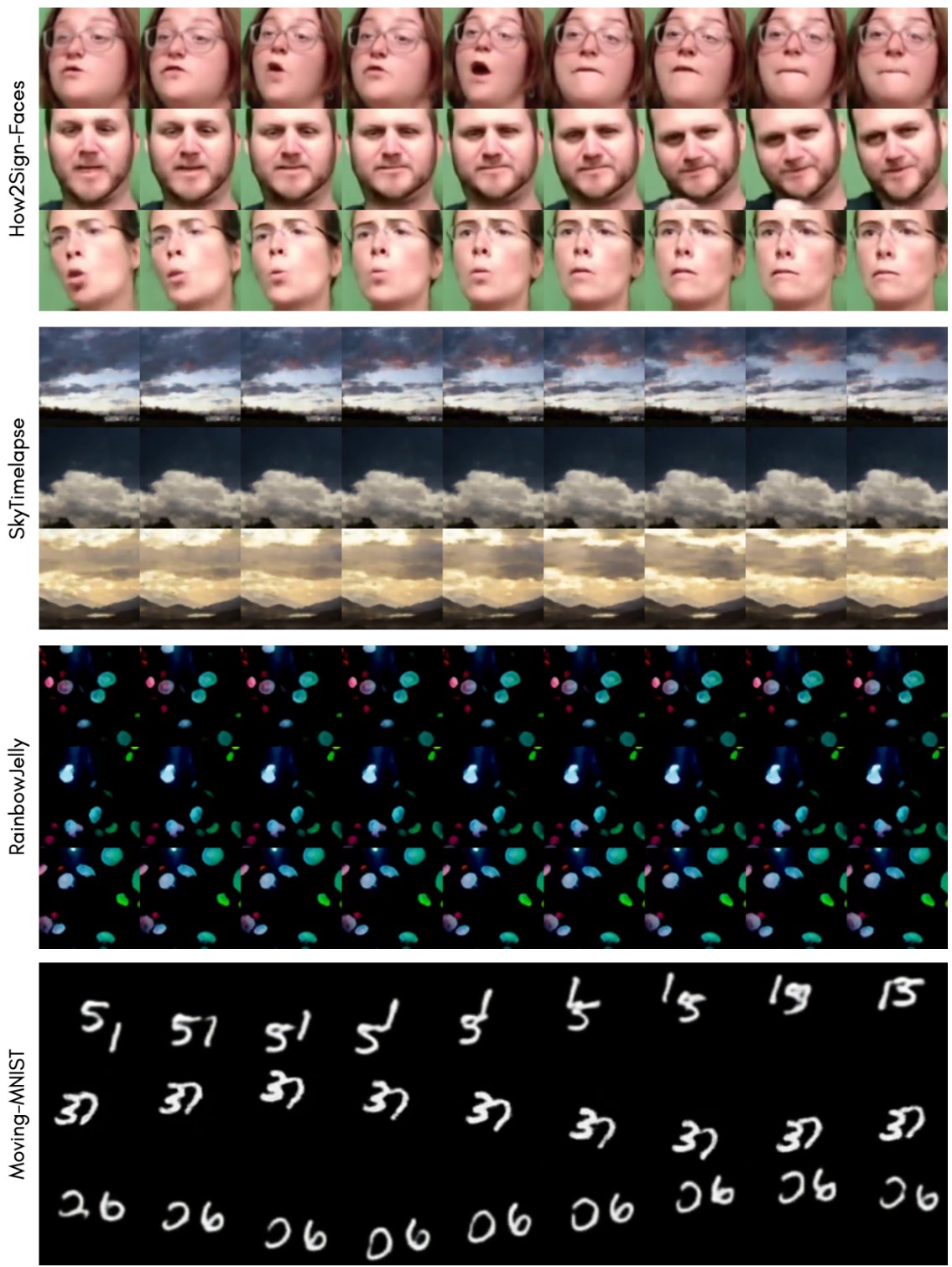

Figure 22: Examples of random videos generated by INR-V.

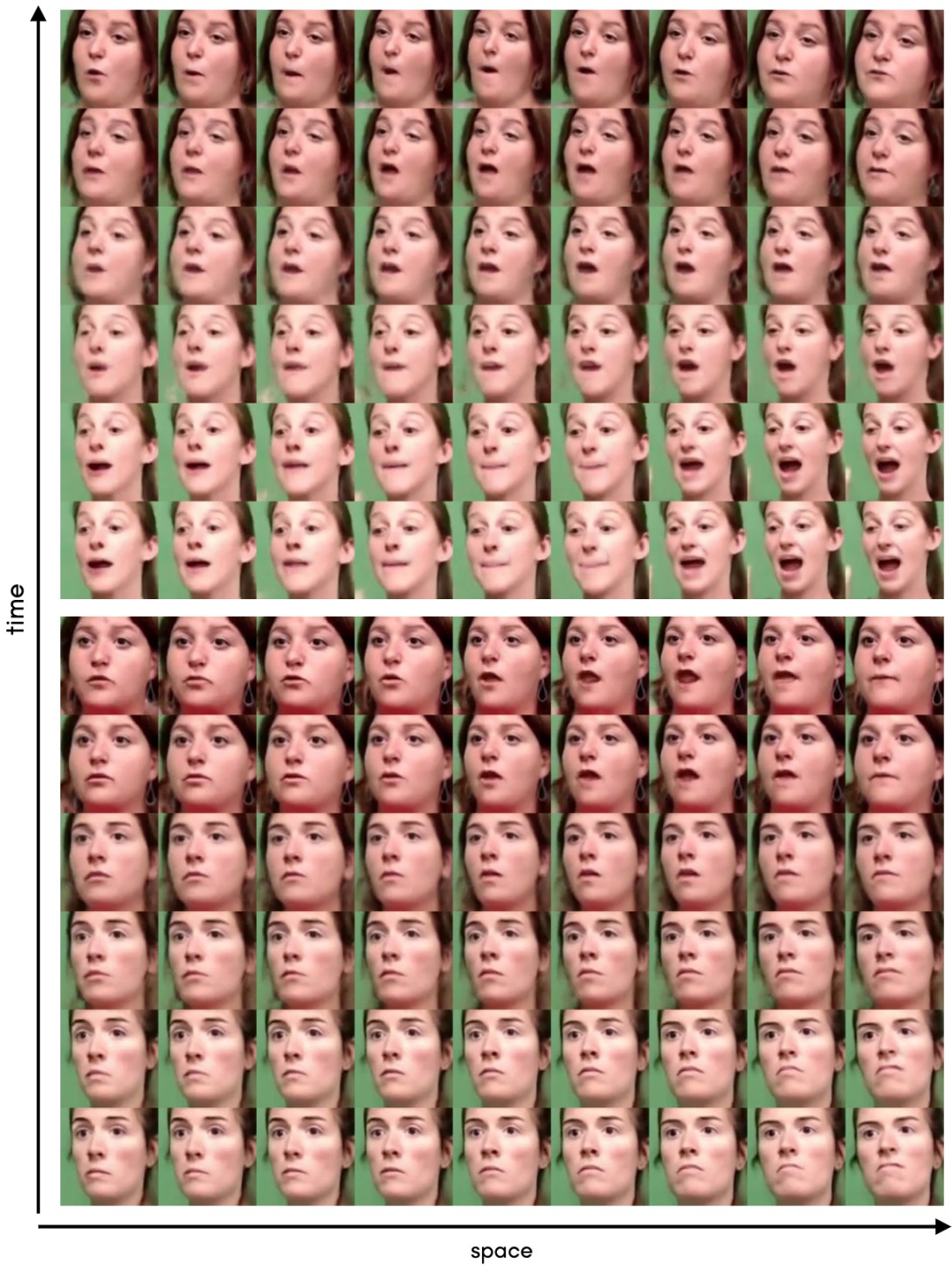

Figure 23: Examples of video interpolation in INR-V on How2Sign-Faces. Two latent points are sampled from the training dataset. Intermediate videos are then generated by sampling intermediate latent points using Slerp interpolation technique. We urge the readers to view the supplementary video for best experience.

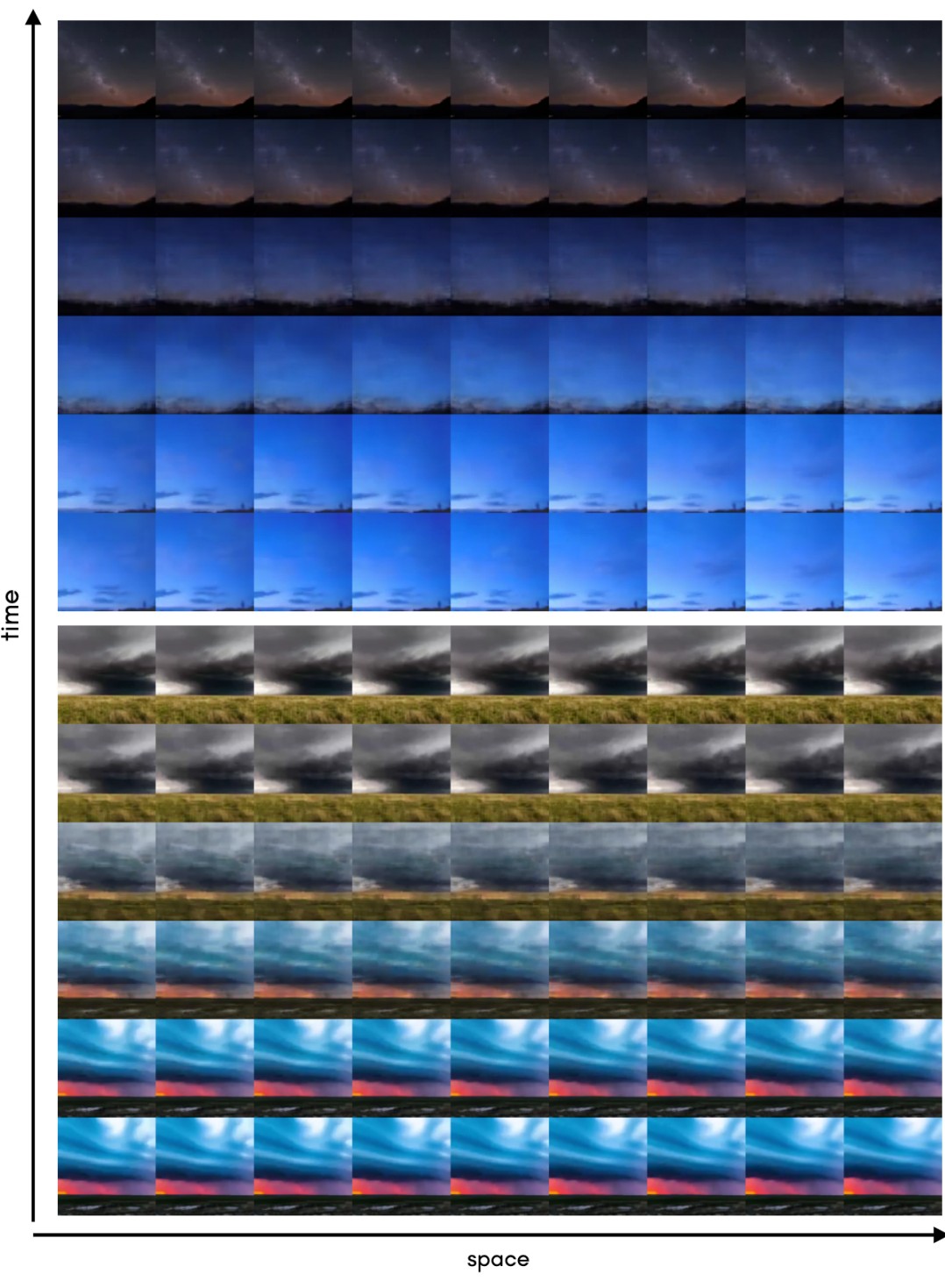

Figure 24: Examples of video interpolation in INR-V on SkyTimelapse. Two latent points are sampled from the training dataset. Intermediate videos are then generated by sampling intermediate latent points using Slerp interpolation technique. We urge the readers to view the supplementary video for best experience.

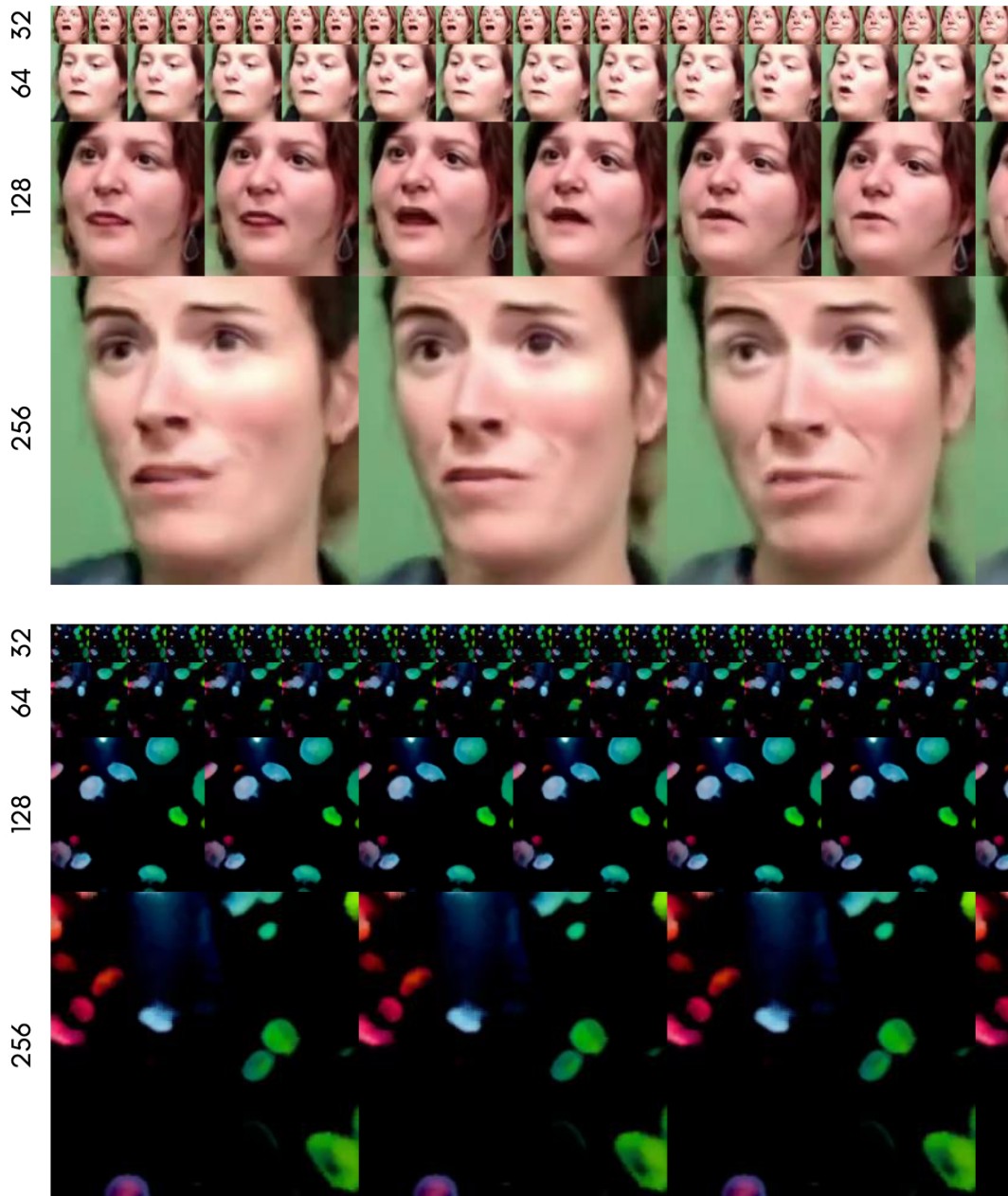

Figure 25: Examples of random videos generated by INR-V at multiple resolutions of $32 \times 32$, $64 \times 64$, $128 \times 128$, and $256 \times 256$ on How2Sign-Faces (top) and RainbowJelly (bottom). The videos are 25 frames long each. The videos are upto scale. INR-V was trained on videos of only $100 \times 100$ resolution. Please refer the supplementary videos for better experience and additional results on 50 frames long video generation.

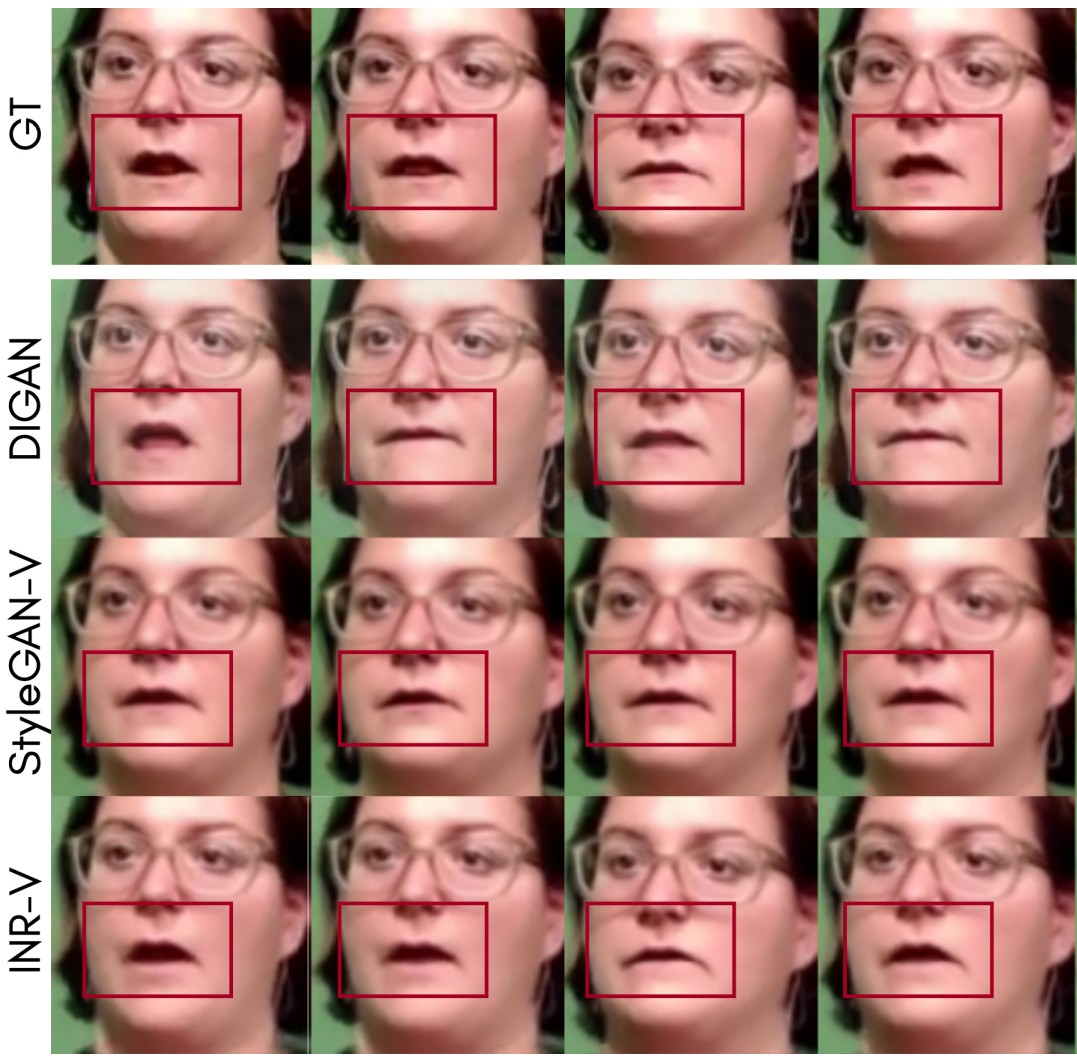

Figure 26: Comparison of video inversion. Red boxes highlight the differences and matches between the ground truth (GT) and the various methods. To note, INR-V is able to preserve the finer mouth movements.

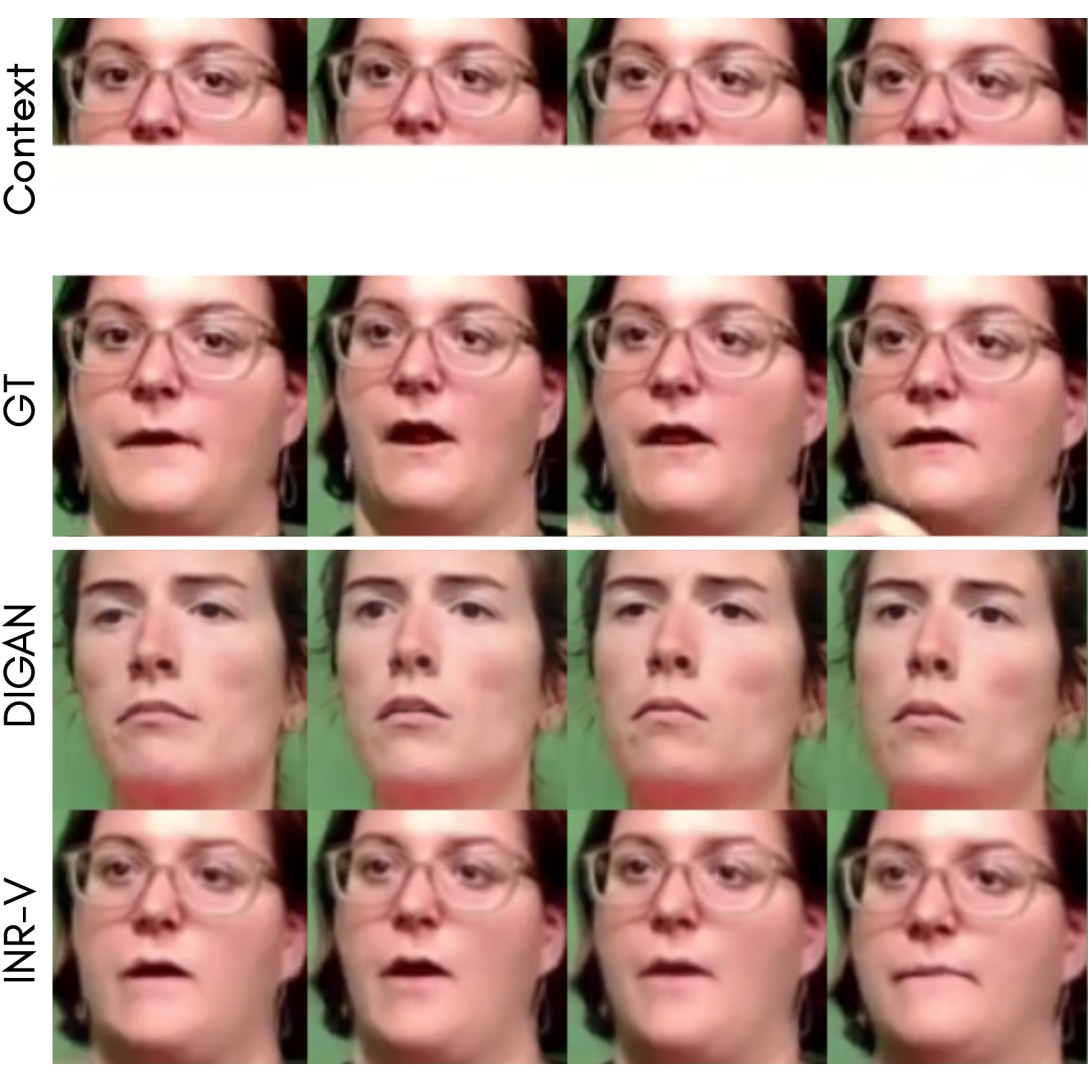

Figure 27: Comparison of half-context inversion in an inpainting setting. At the time of optimization, the model only sees the top half of the video. It then generates the full video back. There can be multiple correct predictions, we showcase one such prediction.

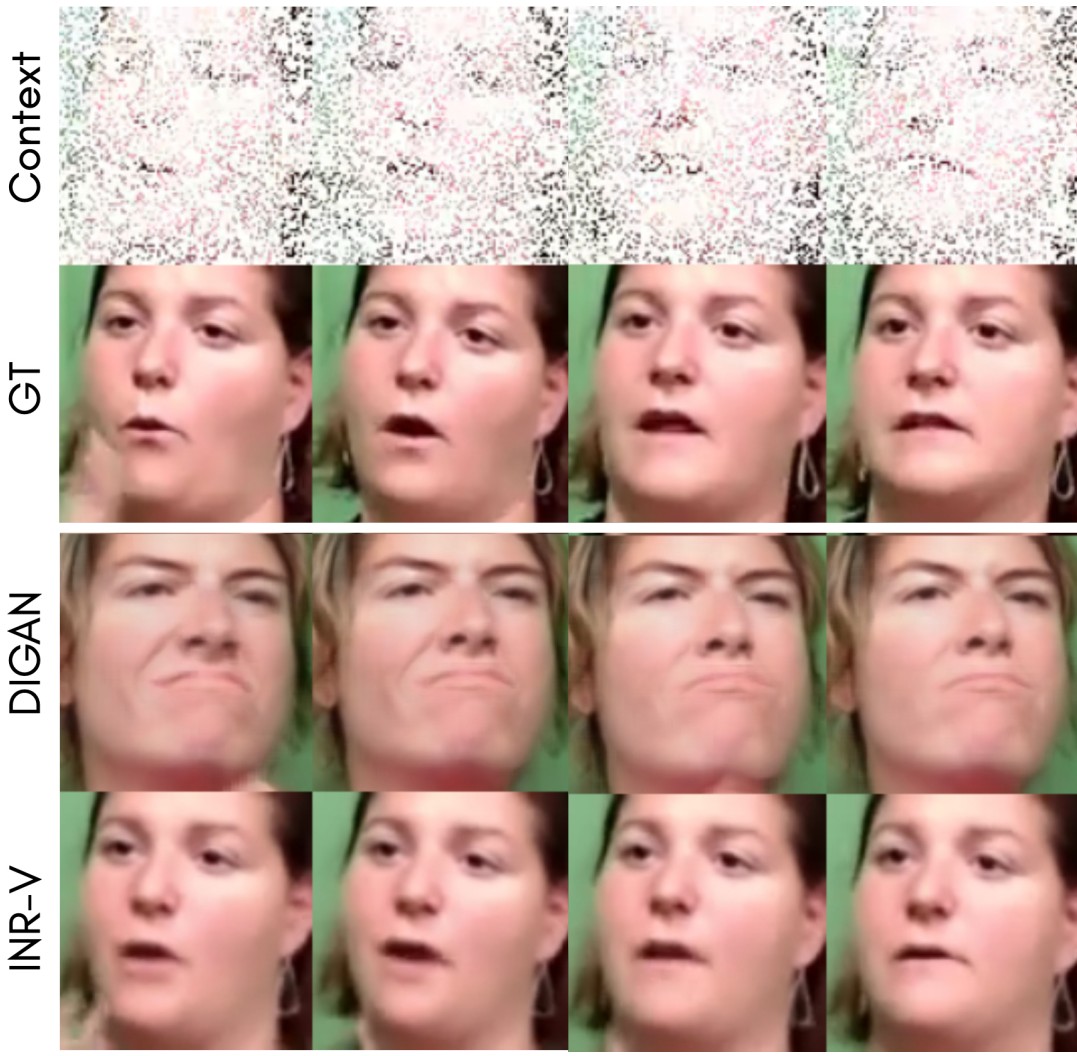

Figure 28: Comparison of half-context inversion in a sparse context setting. At the time of optimization, the model only sees 25% of the full video. INR-V preserve the identity including finer content details like earrings. It also preserves motion like pose and mouth movements.

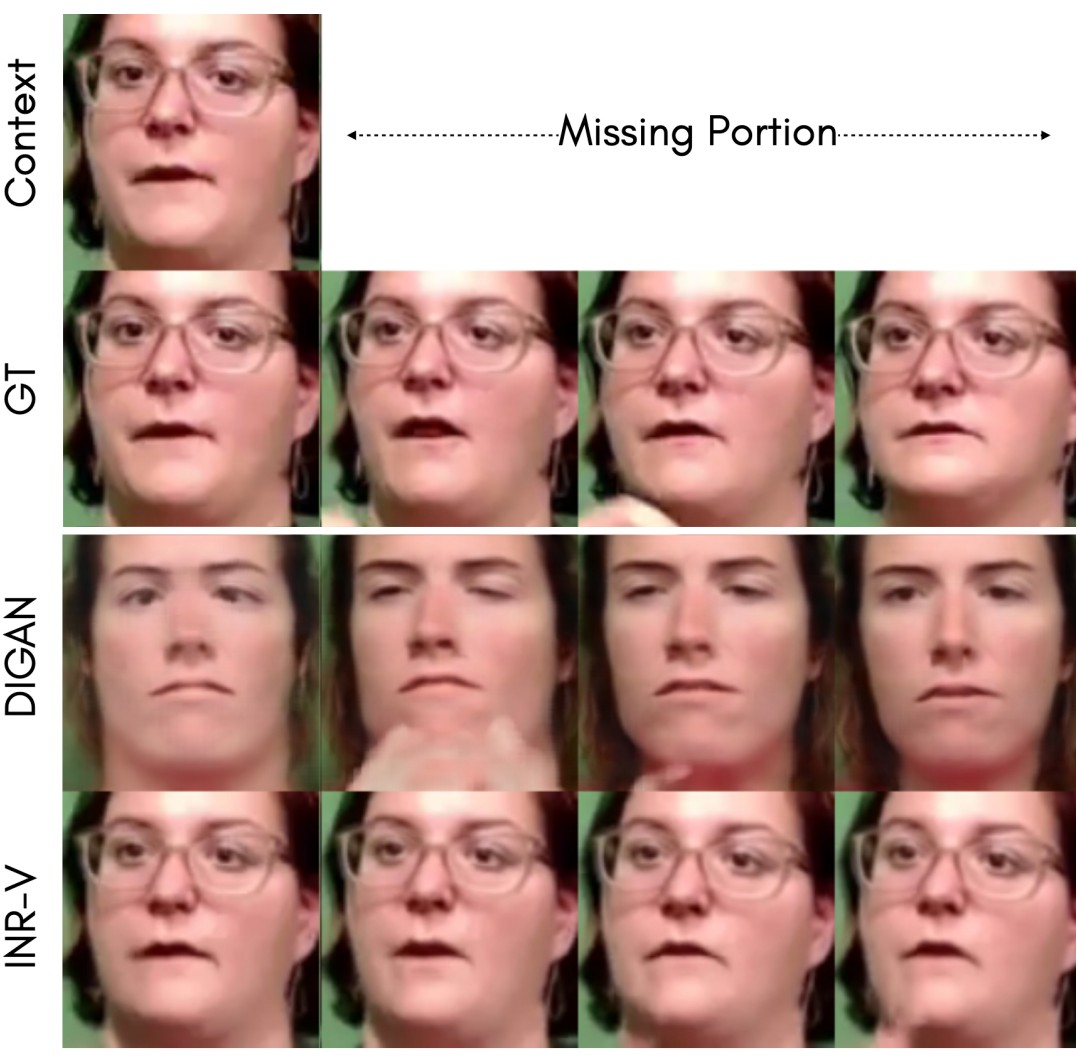

Figure 29: Comparison of half-context inversion in a future frame prediction setting. At the time of optimization, the model only sees the first 4 frames of the video. There can be multiple correct predictions given the identity is preserved across the video. We show one such example.

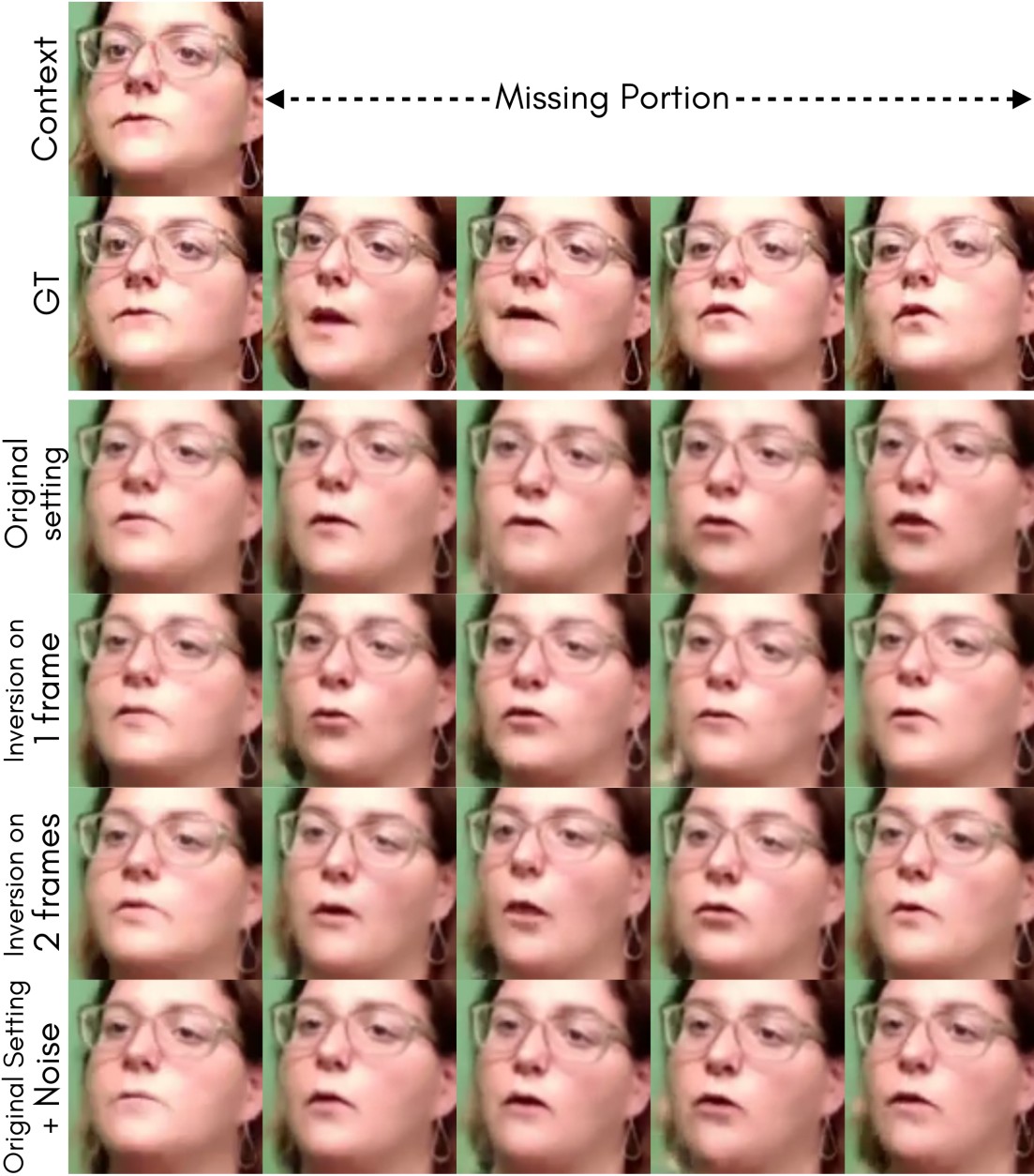

Figure 30: Multimodal Future Frame Prediction: Given the first few frames of a video, we want to predict multiple different outcomes. This can be achieved using INR-V by conditionally optimizing the video's latent vector on half-context. In this example, we condition the latent by varying the number of frames used for inversion or by adding Gaussian noise to the optimized latent. As can be seen, the inversion preserves the seen context; and the future predictions have varying mouth movements, pose, and eye gazes.

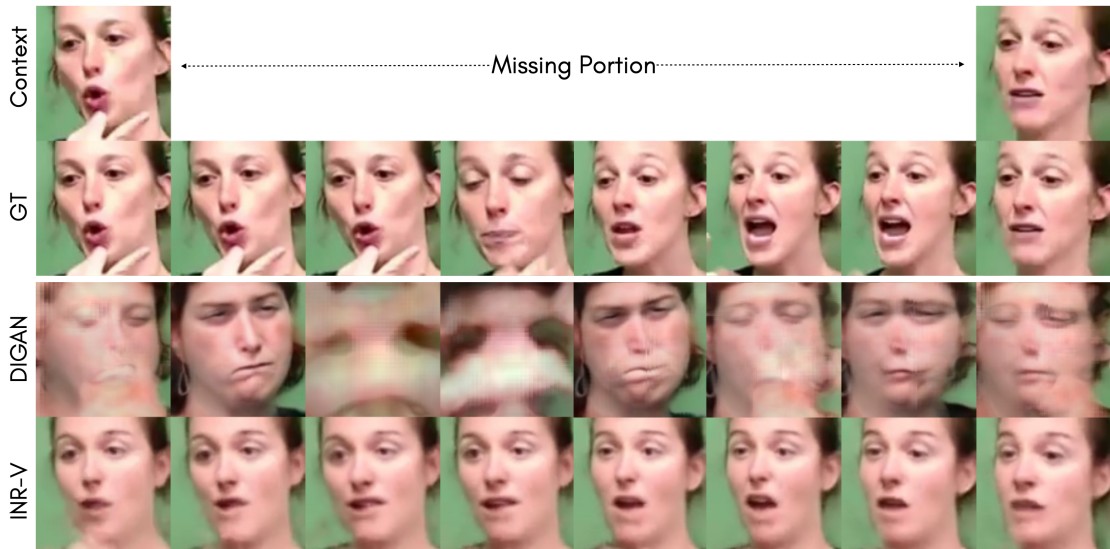

Figure 31: Comparison of half-context inversion in a frame interpolation setting. At the time of optimization, the model only sees the first and last frames of the video. As can be seen, the first and the last frame generated by INR-V match the context (pose, identity, mouth movements), whereas the intermediate frames are very different from the ground truth.

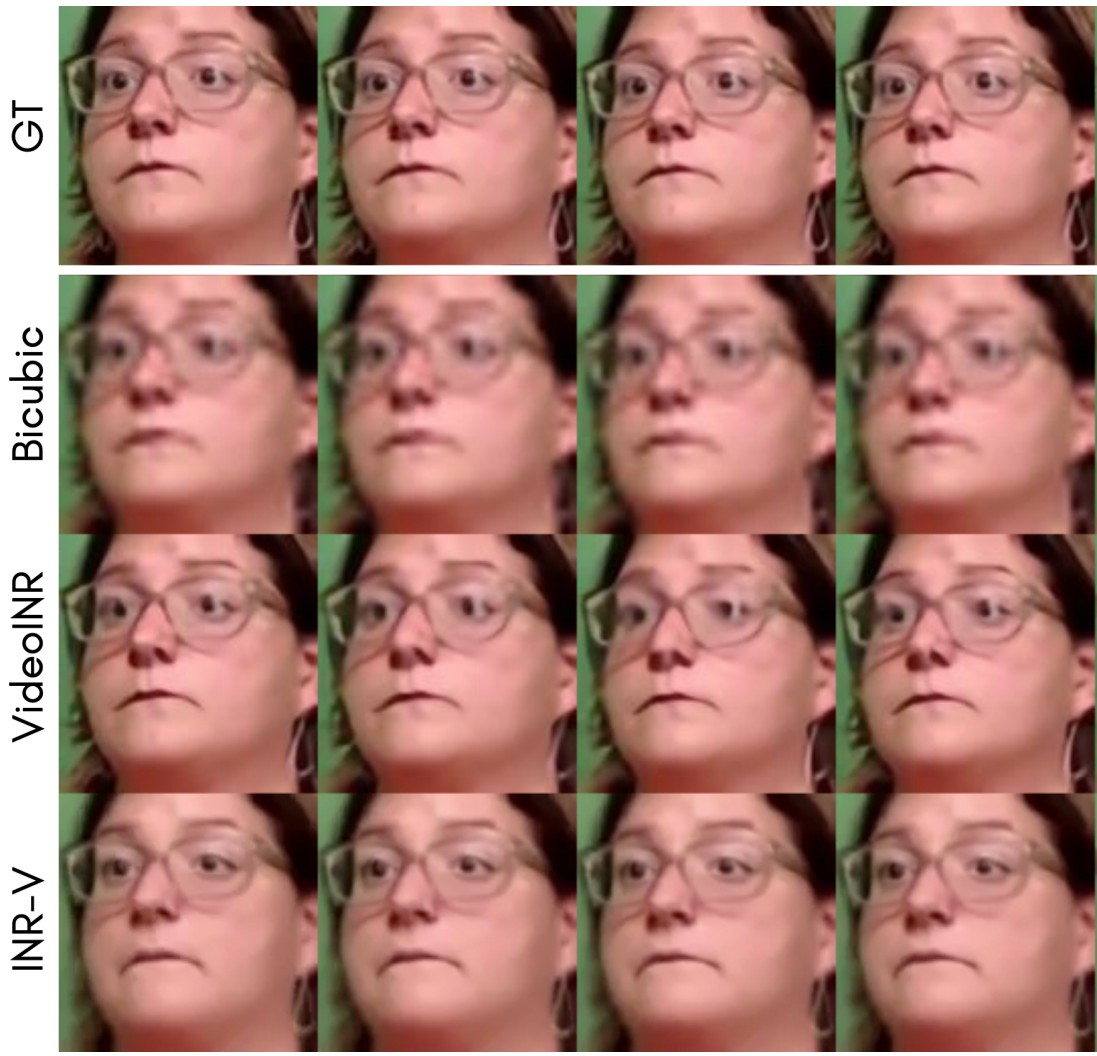

Figure 32: Comparison of video superresolution. A video of $32 \times 32$ is given as input to INR-V for optimization. Once the video is optimized, INR-V regenerates the video at a higher resolution of $128 \times 128$. VideoINR and Bicubic directly see the $32 \times 32$ video and superresolves it to $128 \times 128$. Here, INR-V is not influenced by the glaze on the spectacles and superresolves to a higher dimension closer to the ground truth.

