# OpenReview forum: "INR-V: A Continuous Representation Space for Video-based Generative Tasks"
_TMLR — Accepted by TMLR_

### Review · Reviewer_Ec91 · 2022-07-18

**Summary Of Contributions:**

The main idea of this paper is to represent a video with a NN that can map from a (h,w,t) point to a RGB pixel value for that point. This NN has very few weights, which are dynamically predicted/created by another NN for each video. This is a different video representation from existing methods, and is interesting to study. The main difference from previous INR style works is that the weights are predicted by another network.

The approach is evaluated on video generation tasks, on the How2Sign Faces, SkyTimelapses, Moving-MNIST and RainbowJelly datasets.

**Broader Impact Concerns:**

The limitations and broader impacts seem sufficient.

**Requested Changes:**

Adding the ablation experiments would help the most.

Minor comments:
The abstract is a bit confusing. After reading it, I was unsure of what the paper was proposing. Specifically,
"Images are considered a complete signal, whereas videos are usually broken down into a
set of temporally coherent images. Consequently, an image space is leveraged for various
video-based tasks, "

Is a bit confusing. It'd help to strengthen the idea of the paper to better explain this idea, e.g., why treating a video as a set of frames is bad and what the alternative is.

The INR and meta-network isn't really introduced, and there's no mention of the evaluation tasks (e.g., classification, detection, generation, etc). It'd be good to revise the abstract to give a more clear high-level overview and a mention what tasks this is used for.

The YouTube link in footnote 1 is broken.

**Strengths And Weaknesses:**

The video representation as a NN mapping from location to RGB value is interesting. However, the current approach is only designed for video generation. It isn't immediately clear how or if this would be used for other tasks, like video classification. It would be interesting, and further strengthen the paper, to have a way to use this for those tasks. Otherwise the title seems a bit too broad.


For the key elements:
(1) MLPs lead to fewer parameters than convolutions. I'm not sure this is true? A similar architecture with conv layers could be used with the same number of params? It isn't the choice of MLP vs. convs that leads to this reduction, but rather the configuration (channels, etc) of the MLP.


The experiments compare to other approaches, but this approach relies on a pretrained Video-CLIP model, which uses far more data than the previous works. This makes it a bit unclear how much benefit is coming from that additional data vs. the ideas proposed in this paper.

Related, there aren't any studies on the components of the proposed approach. E.g., effects of video clip, size of codebook, regularization choices, etc. It would strengthen the paper to have more thorough ablation experiments of the components.

---

> ### Author Response · Authors · 2022-09-05
> **Reply to reviewer Ec91**
>
> Thank you for the thoughful review. We have added the revised version of the paper along with the following response.
>
> **The video representation as a NN … the title seems a bit too broad.**
>
> Thank you for pointing this out. We have revised the title of our paper to: *“INR-V: A continuous representation space for video-based generative tasks.”* We believe the diverse tasks we showcase in the paper, such as video inversion, video interpolation, and novel video generation, justify the revised name. We are happy to address any additional concerns regarding the title.
>
> **For the key elements: (1) MLPs … the configuration (channels, etc) of the MLP.**
>
> We believe this comment is made w.r.t to the lines quoted from the introduction section:
>
> *Its INR is free of any convolutional layers and relies on traditional multi-layered perceptrons (MLPs), leading to very few parameters (a few thousand) when compared to the existing SOTA architectures (millions of parameters) Skorokhodov et al. (2021); Tian et al. (2021)*.
>
> Here, our intention is not to claim that MLPs lead to fewer parameters than convolutional layers in general; our claim is specific to the SOTA architectures Skorokhodov et al. (2021); Tian et al. (2021) that are based on convolutional layers. We have rewritten the lines as:
>
> *Its INR is free of expensive convolutional layers (millions of parameters) such as in the existing architectures Skorokhodov et al. (2021); Tian et al. (2021) and relies on a few layers of traditional multi-layered perceptrons (MLPs), leading to a very few parameters (a few thousand)*
>
> **The experiments … the ideas proposed in this paper.**
>
> In Fig. 10, appendix A.1.1, we have added the convergence graph (FVD scores vs. training time) for different training schemes: with/without CLIP regularization and with/without progressive training on two datasets. INR-V converges to the same performance in every setting, taking much longer to converge without regularization or progressive training. We have also visualized the latent codes learned with/without CLIP in Fig. 13. In our network, we compute $m_n$ (input to our hypernetwork $d_\Omega$) by concatenating $g_n$ (computed using VideoCLIP) and $c_n$ (learnable video instance code) and passing them through a neural network $\phi$. In this formulation, the CLIP embeddings used to compute $g_n$ are kept frozen during training; thus, $c_n$ is the only learnable parameter. Therefore, $g_n$ acts as a regularizer for $c_n$ by helping it learn a semantically meaningful space. Without $g_n$, $m_n$ becomes equal to $c_n$, which results in the model learning tight spaces resulting in a mode collapse at the beginning of training and thus taking longer to converge (visualized in Fig. 10 Appendix A.1.1). We have also revised Section 3.2 for added clarity.
>
> **Related, there aren't any studies … of the components.**
>
> We have added the requested ablations in Appendix A.1.
>
> 1. We have showcased the effect of the video clip in A.1.1 by plotting convergence graphs (Fig. 10).
> 2. We have added the performance of INR-V on varying codebook sizes (number of video instances used for training the network) in Appendix A.1.2 (Fig. 12).
> 3. For regularization choices, we have added the performance of INR-V with *Gaussian Regularization* on two datasets along with *No Regularization* and *CLIP Regularization* in Appendix A.1.1 (Fig. 11).
> 4. We have also provided additional insights into the latent space learned by INR-V in Appendix A.2 through (1) t-SNE visualizations on the RainbowJelly dataset (Fig. 13) and (2) a toy dataset “a bouncing ball” with a predefined structure (Fig. 15) to find out if INR-V can learn this structure (Fig. 14).
>
> **Minor comments: The abstract is a bit confusing. … tasks this is used for.**
>
> Thank you for pointing out the limitations in the abstract; we have taken the feedback and updated the abstract accordingly. We hope the abstract has more clarity. We would be happy to incorporate any additional suggestions regarding the abstract.
>
> **The YouTube link in footnote 1 is broken.**
>
> We borrowed the dataset link from StyleGAN-V’s GitHub repository. We will raise an issue on their official website regarding the dataset link and update our paper accordingly.

---

### Review · Reviewer_vVxg · 2022-07-21

**Summary Of Contributions:**

This paper proposes INR-V, a generative model of videos which is based on implicit neural representation. INR-V parametrizes a neural implicit function using a hyper-network. The hyper-network takes a video latent representation and outputs the parameters value of the implicit function. The implicit function is then used to generate the video RGB values.

The authors propose two tricks to stabilize the training of the hyper-network, latent regularization and progressive training (i.e. gradually increasing the size of the training set).

Authors demonstrate the ability of INR-V on various task including, video reconstruction, video generation, interpolation, and various input completion tasks


**Broader Impact Concerns:**

No specific concerns given the broader impact discussion in the paper

**Requested Changes:**

- Improve the clarity of the paper (see previous section).
- Add video generation experiments on UCF-101 and Kinetics.
- Compare computation complexity of INV and convolution networks for video generation.
- Shows that INV-R is working without a CLIP network or better motivate the use of CLIP in the approach.

**Strengths And Weaknesses:**

*Strengths:*
INV-R is interesting, and authors demonstrate  the model flexibility in the empirical evaluation. INR-V can be adapted to solve various tasks, from video generation to future frame prediction and super-resolutions

*Weaknesses:*
While the paper proposes an interesting and novel model for video generation, there are some limitations that need to be fixed before acceptance
- Clarity:
The paper clarity could be improved. In particular
1) What is the code ‘c_n’ and the auto-encoding loss in section 3.2? Why does it help prevent collapse of m_z
2) Why does the representation m_z collapse as this would hurt the reconstruction performance?
3) Are  the hyper-network and INR trained jointly or in different steps?
4) How do you generate new videos in section 4.2 as you don’t train an explicit prior model on the latent space?

- Computational costs of INR-V:
INR-V claims that it can generate large video, while 3D convolution operations would have drastic computational cost. However, implicit functions also have a high computational as you need to call the function for each voxel in your video volume. It would be informative to compare the computational complexity of the two approaches.

- Motivation for the use of CLIP:
How would INR-V perform without using a pre-trained CLIP network. The use of the CLIP seems orthogonal to the main claims. It also provide an advantage to INR-V as CLIP is trained on more data and require more computation.

- INR-V without upsampling:
What is the advantage of INV-R over other video generation approaches as it perform similarly without the upsampling stage.

- Lack of complexity in the evaluation datasets:
All the datasets considered for evaluation have limited visual complexity and almost no camera motion. How would INV-R perform on more complex dataset such as UCF-101 and Kinetics?


Additional question:
- Video prediction is a multimodal prediction tasks as several outcomes are often plausible for a given input.  How is INV-R able to generate different futures for the same set of input frames?

---

> ### Author Response · Authors · 2022-09-05
> **Reply to reviewer vVxg**
>
> Thank you for your insightful review. We have also added a revised version of the paper, along with our response below.
>
> 1. **Clarity:The paper clarity could be improved. In particular**
>     1. **What is the code … of m_z**
>
>         In our network, we compute $m_n$ (input to our hypernetwork $d_\Omega$) by concatenating $g_n$ (computed using VideoCLIP) and $c_n$ (learnable video instance code) and passing them through a neural network $\phi$. In this formulation, the CLIP embeddings that are used to compute $g_n$ are kept frozen during training; thus, $c_n$ is the only learnable parameter. $c_n$ is modified in an auto-decoding fashion during training. The auto-encoding loss is computed on $g_n$; however, since the CLIP embeddings are non-learnable, only the GRU component of Video-CLIP changes. Thus, $g_n$ acts as a regularizer for $c_n$ by helping it learn a semantically meaningful space. Without $g_n$, $m_n$ becomes equal to $c_n$, which results in the model learning tight spaces resulting in a mode collapse at the beginning of training and thus taking longer to converge (visualized in Fig. 10 Appendix A.1.1). We have also revised Section 3.2 for added clarity.
>
>     2. **Why does the representation … reconstruction performance?**
>
>         Without added regularization, at the start of the training, the video latents are formed in a condensed space with no semantic meaning. This is because $m_n$ is equal to $c_n$ and without a semantic encoder like CLIP, $m_n$ is initialized around a point (Fig. 13 top left). As a result, the model struggles to learn a space suitable for large and diverse datasets like How2Sign-Faces or RainbowJelly resulting in a mode collapse during the initial phase of training (visualized qualitatively in Fig. 9).  To accommodate the reconstruction loss, the model eventually converges but takes a long time to do so (Fig. 10).
>
>     3. **Are the hyper-network and INR trained jointly or in different steps?**
>
>         The entire network is trained jointly. Please refer to Section 3.1, Page 5 “At the time of training, $\Omega$ and $m_n$ are optimized together.” We have also revised Section 3 for improved clarity.
>
>     4. **How do you generate new … on the latent space?**
>
>         In our formulation, the hypernetwork learns a mapping from an underlying meta-distribution tau to a valid neural representation space by getting trained on multiple valid neural representations. This technique has been used in recent works like Light Field Networks Sitzmann et al. (2021), DeepSDF Park et al. (2019), Occupancy Networks Mescheder et al. (2018), Continuous 3D-Structure-Aware Neural Scene Representations Siren Sitzmann et al. (2020a), MetaSDF Siren Sitzmann et al. (2020b), and SIREN Sitzmann et al. (2020). To sample novel videos, tau can be queried in different ways. We sample novel videos by interpolating latent points between known latent points (video instances seen during training) through Slerp interpolation. We also show conditional sampling in Section 5.3 (Video Superresolution through Inversion), where 32 $\times$ 32-dimensional unseen video instances are used to conditionally sample the latent points from tau for generating higher resolution videos. The learned space is not dense but supports the ability to generate diverse and novel video instances. INR-V’s formulation itself does not prevent a dense space, and the prior can be conditioned to become denser. For instance, we show Gaussian regularization over the prior in Fig. 11, Appendix A.1. Additional insights about the learned latent space are added in Appendix A.1 and A.2.
>
> 2. **Computational costs of INR-V: … complexity of the two approaches.**
>
>     To analyze the computation overhead of INR-V, we will plot the time taken to generate a single video against the dimension of the video in the next iteration of the paper on (1) INR-V and (2) a 3D convolution-based video generation model. The computational complexity can be divided into time and memory. In terms of time, despite needing to call the function for each pixel of our video, we can generate the entire video (up to 256 $\times$ 256, 25 frames long) in a single forward pass of INR-V on a single GPU of NVIDIA GTX 2080 ti by batching all the pixels of the video. In terms of memory, since INR-V predicts a video one pixel at a time, it can predict an arbitrarily large or long video by predicting the video in phases. A 3D Conv network, however, would be constrained by the hardware memory and would fail to generate an arbitrarily long/large video. We will show this comparison quantitatively in the next iteration of the paper.

---

> > ### Author Response · Authors · 2022-09-05
> > **Reply to reviewer vVxg (continued)**
> >
> > 3. **Motivation for the use of CLIP: How … and require more computation.**
> >
> >     In appendix A.1.1, we have added the convergence graph (FVD scores vs. training time) for different training schemes: with/without CLIP regularization and with/without progressive training on two datasets (Fig. 10). INR-V converges to the same performance in every setting taking much longer to converge without regularization or progressive training. We have also visualized the latent codes learned with/without CLIP in Fig. 13.
> >
> > 4. **INR-V without upsampling: What is the advantage … without the upsampling stage.**
> >
> >     Unlike existing image-based video generator networks, INR-V possesses a unique property of sampling novel videos through interpolation. One can sample intermediate points between known latent points using existing interpolation techniques and expect the generated videos to have spatial and temporal coherence and similarity to the known points. This can be useful for learning a structure in continuous and semantically meaningful datasets. We showcase the effects of such a learned space through continuous interpolation on a toy dataset comprising 50 videos of “a bouncing ball” in Section A.2.2. We also show this phenomenon in Fig. 6, Fig. 22, and Fig. 23 of the main paper and timestep 5:30 - 6:15 of the supplementary video. We can clearly observe a smooth transition in the person’s pose, identity, mouth movements, and expressions in How2Sign-Faces and a smooth transition in the sky's color and the clouds' position in the SkyTimelapse dataset. Additionally, INR-V outperforms the existing baselines on many diverse generative tasks - video inversion, video interpolation, frame prediction, video inpainting, and unlike existing baselines, can perform the task of video compression.
> >
> > 5. **Lack of complexity in the evaluation datasets …**
> >
> >     Due to extreme diversity and limited structure in action recognition datasets like UCF-101 and Kinetics, INR-V cannot learn a meaningful representation space. However, to showcase that INR-V can learn a meaningful space on structured datasets with considerable camera motion and subject movement, we train INR-V and the existing baselines on a single class of the UCF-101 dataset — JumpRope that has considerable camera motion and subject movement. We also train our model and the existing baselines on two additional datasets, Taichi [[https://proceedings.neurips.cc/paper/2019/hash/31c0b36aef265d9221af80872ceb62f9-Abstract.html](https://proceedings.neurips.cc/paper/2019/hash/31c0b36aef265d9221af80872ceb62f9-Abstract.html)] and BAIR [[http://rail.eecs.berkeley.edu/models/savp/tables/bair_all/](https://rail.eecs.berkeley.edu/models/savp/tables/bair_all/)], with considerable motion. Please find the comparisons below in terms of FVD scores (lower the better).
> >
> >                       ------------------------------------------------------------
> >                       |                 |  TAI-CHI  |  BAIR   |   UCF-JumpRope   |
> >                       ------------------------------------------------------------
> >                       |  MoCoGAN-HD     |   677.83  |   795   |     1385.85      |
> >                       |  DIGAN          |   324.12  |   316   |      814.28      |
> >                       |  StyleGAN-V     |   418.96  |   432   |     1014.24      |
> >                       |  INR-V          |   382.21  |   458   |      756.45      |
> >                       ------------------------------------------------------------
> >
> > **Additional question:**
> >
> > 1. **Video prediction is a multimodal prediction tasks as several outcomes are often plausible for a given input. How is INV-R able to generate different futures for the same set of input frames?**
> >
> >     We address this in two different ways: (1) conditioning the generation using random Gaussian noise and (2) inverting on varying amounts of context — choosing to use a random number of frames from the given video segment (for instance, any 3 out of the 4 frames in the given segment). We will add the results of this conditioning in the subsequent iteration of the paper.

---

### Review · Reviewer_G6uQ · 2022-08-26

**Summary Of Contributions:**

The submitted work presents a method for learning a continuous representation space specifically for videos. The space is continuous in that the latent codes represent whole videos and not frame-by-frame instances. Being a continuous representation space, the model can more easily generate videos that are coherent in both motion and content without sudden and unrealistic changes. For this, crucial part plays the use of pretrained CLIP encoders, without which the method appears to suffer. Applications of this method include video generation, inpainting, future prediction, and so on (or what the method succinctly calls ‘video inversion’). Other contributions claimed by the paper are having a neural representation for video that is in the order of thousands instead of millions, as well as a suggested training methodology to stabilize the training of the notoriously unstable hyper networks.


**Broader Impact Concerns:**

No concerns.

**Requested Changes:**

See list of weaknesses.

**Strengths And Weaknesses:**

Positives

I am very sympathetic to all objectives and desiderata put forward by the paper. Learning a representation space that is tailored to spatiotemporal data and video is certainly missing in the literature. A simplistic view of merely extending existing image-based representation learning methods is not sufficient, as also demonstrated in the submission.

I believe that this work follows the late trend of using the so called ‘foundational models’ (such as CLIP) for secondary tasks. The results show that indeed CLIP can play a critical role to learning of good video representations.

The visualizations and results look quite nice at times, and this shows that the method is onto something.

Negatives

1. Despite being sympathetic to the work and its objectives, unfortunately it is not clear to me what is the message that the submission is striving for.

Is the message ‘this is a nice new representation space for videos’? If yes, I would expect deeper analysis with respect to downstream tasks that are supposed to be benefit from a representation space (action classification, event classification, …), together with appropriate comparisons against relevant supervised/unsupervised/self-supervised baselines. What is more, I am doubting whether one can make such a claim when the method cannot work without using literally and explicitly and external method (CLIP) with its trained weights etc. If anything, CLIP is image-based, so at least the claims regarding video specific representations should be toned down.

Is the message ‘Using CLIP (or similar foundational models) is critical for future works on the video representation space’? I assume not, since this is not emphasized in the contributions. However, without CLIP I doubt whether the suggested method would still work well enough, as also demonstrated in ablation A.1.

2. The paper does not have clear contributions from a technical point of view, and the overall method is mostly a combination of existing and well known techniques which are expected to work well together, one way or the other. The method can basically described as
    1. We represent videos as functions whose parameters are learned to be predicted by hypernetworks.
    2. To avoid collapse of the latent space to the observed points, the ‘latent’ codes are a combination of CLIP codes and autoencoding codes. To avoid collapse, also, bootstrapping must be used.
    3. Because the generations do not look always that crisp, we also use an additional denoising step borrowed from VQVAE.
3. Going deeper into the technical aspects, I am not even sure whether the method is correct. At several locations in section 3.1, it is mentioned that a video is represented as a theta-parameterized function, sampled from tau. Also, it is mentioned that the representations are constrained on tau.
    1. To sample from any distribution, one must have a probability density function either explicitly (like with VAEs) or with implicitly (like GANs). Like that, one can use basic samplers (like for gaussians in VAE), or advanced samplers that rely on basic samplers (like the decoder of a GAN or of a Normalizing Flow). How are samples sampled in this paper, especially given that CLIP is not generative as far as I know? Are samples obtained ad hoc via a uniform sampling? If yes, why does this work and how can one be sure that indeed the points in learned latent space are indeed distributed uniformly, that there are no ‘empty spaces’, etc? This is even more worrisome given that the method claims to be able to generate novel samples without explaining how this is possible in the training (only an L2 loss is described, which alone simply guarantees optimization/fitting).
    2. It is quite unclear to me what is ‘constraining a space’? Is the space shaped according to a particular topology? Or is it meant that only part of the CLIP space is really considered, so placing an ad hoc constraint. The term regularization or constraint has usually a specific (and different) meaning.
    3. The progressive training of section 3.2 seems ad hoc and thus sensitive to the way it is defined or implemented. For instance, how are the 10 examples selected that kick start the training, and how sensitive is the method to them? And what about the additional examples included at each round? Furthermore, I suppose the mini-batching takes places within the set V? Otherwise how does the mini batches fit to memory?
4. While the results are not bad, they are not that convincing either. In terms of metrics, the method appears to be doing well natively on Rainbow Jelly, and about the same on Moving MNIST (table 1). With the extra denoising step the model does better than baselines, however, it is not clear if this denoising step would also benefit the baselines (in that sense, perhaps it makes sense to more tightly integrate the denoising into the model).
5. More generally, it is no clear if all comparisons are all that fair, given that the proposed method relies on an external component (CLIP) that could easily be plugged to the other baselines. For instance, if CLIP was combined with the GAN baselines, would the sudden irregularities in figure 6 still be there?
6. There are several passages that are quite unclear because terminology is assumed to be known, which however will not necessarily be the case.
    1. In 5.1 please explain what is Slerp interpolation.
    2. There is significant notation clutter in 3.1 with new notations used only once or twice.
    3. In the beginning of 5.2 it is really unclear to me what the method is supposed to do, and how can it work with different lengths and scales.
    4. Perhaps it makes sense to explain briefly what it is meant by ‘video inversion’ in the beginning of 5.3. It becomes later on clear in the text, however, not all readers might be familiar with the term.
    5. Table 4 is inconsistent compared to previous tables. There is no bold face, nor arrows indicating when the number is better.
    6. I am not sure if one can use the word ‘superresolve’, although I guess it is technically correct.
    7. Also, the section in ‘Quantitative evaluation’ in 5.3 is very unstructured and it is hard to understand what it means/follow.
7. A last point, which however I do not consider that important, is that continuous representations (in the sense of a representation from the entirety of a video) were previously considered as well, in the context of action recognition. An example is Fernando et al., Modeling Video Evolution for Action Recognition, CVPR 2015, or Action Recognition with Dynamic Images, which learn to represent a video as the parameters of a linear SVR or an approximation thereof (in a deep network setting).

---

> ### Author Response · Authors · 2022-09-05
> **Reply to reviewer G6uQ**
>
> Thank you for your detailed review. Along with the following response, we have also added a revised version of the paper.
>
> 1. **Despite being sympathetic … together with appropriate comparisons against relevant supervised/unsupervised/self-supervised baselines.**
>
>     Our motivation is to propose an improved representation space for video-based generative tasks. We evaluated this space through quantitative and qualitative comparisons on several downstream tasks — video inversion, novel video generation, video interpolation, video inpainting, and future-frame prediction. To better highlight our motivation and the representation space learned by such a network, we have revised the title of our paper from “*INR-V: A continuous representation space for videos*” to “*INR-V: A continuous representation space for video-based generative tasks*”
>
>     **What is more, I am doubting whether one can make such a claim … still work well enough, as also demonstrated in ablation A.1.**
>
>     Thank you for raising this point. To clarify, the method can work without CLIP regularization. We have addressed this point in the revised version. Specifically, we have added a complete convergence comparison for the various setting: with/without CLIP regularization, with/without progressive training, on two datasets in Appendix A.1; the convergence graphs clearly show that INR-V without any regularization converges to the same FVD value as with CLIP regularization or additional progressive training while taking a much longer time to converge (Fig. 10). We have also added latent visualization in Appendix A.2 (Fig. 13), which clearly indicates that CLIP helps in learning a rich prior for our task. Thus, the generation capabilities are possible based on the proposed method without resorting to foundational models.
>
> 2. **The paper does not have clear contributions…we also use an additional denoising step borrowed from VQVAE.**
>
>     Our contribution lies in formulating a novel representation space for video-based generative tasks. We extensively and empirically evaluate our network on diverse tasks, including novel tasks that were not possible before with existing architectures (StyleGAN-V, DIGAN, MoCoGAN-HD), such as (1) smooth video interpolation (we refer the reviewer to our supplementary video - timestep - 5:30) and (2) video inversion (timestep 8:15) - inverting a full video to its corresponding latent point in the learned representation space of the pretrained generator. Existing works resort to frame-by-frame video inversion. Consequently, INR-V significantly outperforms them on these tasks. Our contribution also lies in stabilizing the hypernetwork training through CLIP regularization and progressive training. However, we showcase that INR-V works equally well without CLIP regularization, while taking longer to converge in Appendix A.1. We have also revised Section 3.2 to improve the clarity.
>
> 3. **Going deeper into the technical … that the representations are constrained on tau.**
>
>     We have revised Section 3.1. Please revisit and check if the revision clarifies your concerns.
>
>     1. **To sample from any distribution, one must … guarantees optimization/fitting).**
>
>         In our formulation, the hypernetwork learns a mapping from an underlying meta-distribution tau to a valid neural representation space by getting trained on multiple valid neural representations. This technique has been used in recent works like Light Field Networks Sitzmann et al. (2021), DeepSDF Park et al. (2019), Occupancy Networks Mescheder et al. (2018), Continuous 3D-Structure-Aware Neural Scene Representations Siren Sitzmann et al. (2020a), MetaSDF Siren Sitzmann et al. (2020b), and SIREN Sitzmann et al. (2020). To sample novel videos, tau can be queried in different ways. We sample novel videos by interpolating latent points between known latent points (video instances seen during training) through Slerp interpolation. We also demonstrate conditional sampling in Section 5.3 (Video Superresolution through Inversion), where 32 $\times$ 32-dimensional video instances are used to conditionally sample the latent points from tau for generating higher resolution videos. The learned space is not dense but supports the ability to generate diverse and novel video instances. INR-V’s formulation itself does not prevent a dense space, and the prior can be conditioned to become denser. For instance, we show Gaussian regularization over the prior in Appendix A.1 (Fig. 11). Additional insights about the learned latent space are added in Appendix A.1 and A.2.
>
>     2. **It is quite unclear to … meaning.**
>
>         We have revised Section 3.1. Please revisit and check if the revision clarifies your concerns.

---

> > ### Author Response · Authors · 2022-09-05
> > **Reply to reviewer G6uQ (continued)**
> >
> > 3. **The progressive training of … mini batches fit to memory?**
> >
> >     We randomly selected the videos for each stage in progressive training and observed similar performance across the random batches. We will include a convergence graph for showcasing this phenomenon in the next iteration of the paper. Yes, mini batching takes place within set $V$. Each stage in the progressive training is a full training of the network with the weights of the model in the current stage initialized with the weights learned from the previous stage. This includes the learnable video instance codes seen in the previous stage. The new instance codes added during the current stage are initialized from a Gaussian distribution. We have also revised Section 3.3 to include these changes.
> >
> > 4. **While the results are not bad … the denoising into the model).**
> >
> >     The existing works rely on image-based generators. Such generators have seen years of research and have been known to show perceptually high-quality videos. We generate videos without relying on any image generators. There also has been extensive research in the areas of image and video enhancement (Yang et al. (2021); Chu et al. (2020); Liang et al. (2022); Chadha et al. (2020)), and they are a much-more evolved domain with powerful results. Thus, we rely on existing networks to improve our video quality. With VQVAE2, we demonstrate the ease with which the perceptual quality of our network can be improved by training it naively without any architectural changes. We do not claim VQVAE2 as our novelty. We have also listed the perceptual quality as a limitation of INR-V and are excited to see future works that can improve the quality of our network. However, we demonstrate the potential of our network and the proposed representation space through many diverse generative tasks, such as video interpolation and video-inversion-based tasks, where we clearly outperform the existing networks.
> >
> > 5. **More generally, it is no clear … figure 6 still be there?**
> >
> >     We believe that we have addressed this now in our ablations reported in A.1. We are happy to answer any additional concerns regarding the comparison.
> >
> > 6. **There are several … be the case.**
> >     1. **In 5.1 please explain what is Slerp interpolation.**
> >
> >         We have added a footnote to an existing popular article that explains Slerp.
> >
> >     2. **There is significant notation clutter in 3.1 with new notations used only once or twice.**
> >
> >         We have revised this section in the revised version. If there are additional concerns, we are happy to address them.
> >
> >     3. **In the beginning of 5.2 … and scales.**
> >
> >         To generate a video of dimension H x W x T, we sample H, W, and T number of points spaced equally between [-1, 1] as the input to the INR function. The pixel positions h, w, and t vary between -1 and 1. For a higher resolution, we sample points more finely, and for a lower resolution, we sample points sparsely. We have rearranged the section and specified how the dimension is varied.
> >
> >     4. **Perhaps it makes … the term.**
> >
> >         We have added the definition at the beginning of Section 5.3.
> >
> >     5. **Table 4 is … number is better.**
> >
> >         We have added the arrows and bold faces to the table.
> >
> >     6. **I am not sure … is technically correct.**
> >     7. **Also, … /follow.**
> >
> >         We have modified the Section to make it more structured. We are happy to address additional clarifications.
> >
> > 7. **A last point, which however … network setting).**
> >
> >     Thank you for pointing these works out; we have added them to our Related Work section under Video Generation. We hope that with the changed title, this concern has been addressed.

---

> > > ### Comment · Reviewer_G6uQ · 2022-10-03
> > > **Response to reviewers' comments**
> > >
> > > I find all my points by reviewers addressed well enough. The main objective of the work is novel, generally, and the authors do make a good point that prior works specialize on individual tasks. I thus recommend acceptance.

---

### Comment · Action_Editors · 2022-09-05
**author discussion**

Dear reviewers,

the authors have now replied to your reviews. I believe you will be able to submit your recommendations starting from next week, so this week is a good time to check if you have all the information you need (from authors and/or other reviewers).

Thanks again for your valuable time,

Joao

---

### Author Response · Authors · 2022-09-19
**Response to all reviewers and AE**

We sincerely thank and appreciate all the reviewers' and AE's time and efforts in thoroughly reviewing our work and for providing their valuable comments and suggestions. This has helped us strengthen our submission.

With this iteration of the paper, we hope that we have addressed all the concerns of the reviewers through revised sections, additional experiments, ablations, qualitative results, and quantitative metrics, which are marked in blue. We are happy to address additional questions or provide more information.

We again thank the reviewers' and the AE for their time and efforts.

---

### Decision · Action_Editors · 2022-10-13

**Recommendation:** Accept with minor revision

**Comment:**

All reviewers are happy with the improvements made by the authors during the reviewing period and are at least leaning towards acceptance. One of  the reviewers thinks it is still unclear in the paper "if this type of approach could be applied to more complex video datasets", so I would suggest addressing that comment in a minor revision. Otherwise I believe the paper is in a good state and a welcome addition to TMLR.



**Audience:**

Video generation is a hot topic so contributions in this space are likely to attract attention in the TMLR audience.

**Claims And Evidence:**

Yes, all claims are supported by compelling evidence.

---

> ### Author Response · Authors · 2022-10-28
> **Camera-ready Version**
>
> We want to thank all the reviewers and the AE again for their valuable comments and suggestions that have contributed to the camera-ready version of our paper. We have addressed the reviewer's comment on "if this type of approach could be applied to more complex video datasets" in the **Limitations** subsection of Appendix A.6. We have also added the camera-ready supplementary video in this version along with the deanonymized camera-ready paper.
>
> We thank everyone for a smooth and swift review process.